# REOrdering Patches Improves Vision Models

**Declan Kutscher[1]** **David M. Chan[2]** **Yutong Bai[2]** **Trevor Darrell[2]** **Ritwik Gupta[2]**

[1]University of Pittsburgh  [2]University of California, Berkeley

## Abstract

Sequence models such as transformers require inputs to be represented as one-dimensional sequences. In vision, this typically involves flattening images using a fixed row-major (raster-scan) order. While full self-attention is permutation-equivariant, modern long-sequence transformers increasingly rely on architectural approximations that break this invariance and introduce sensitivity to patch ordering. We show that patch order significantly affects model performance in such settings, with simple alternatives like column-major or Hilbert curves yielding notable accuracy shifts. Motivated by this, we propose *REOrder*, a two-stage framework for discovering task-optimal patch orderings. First, we derive an information-theoretic prior by evaluating the compressibility of various patch sequences. Then, we learn a policy over permutations by optimizing a Plackett-Luce policy using REINFORCE. This approach enables efficient learning in a combinatorial permutation space. *REOrder* improves top-1 accuracy over row-major ordering on ImageNet-1K by up to $3.01\%$ and Functional Map of the World by $13.35\%$.

## 1 Introduction

Autoregressive sequence models have become the backbone of leading systems in both language and vision. These models operate on images by first converting their 2-D grid structure into a 1-D sequence of patches. Conventionally, a row-major order is used for this linearization under the assumption that full self-attention is utilized by the model. As self-attention, augmented with positional embeddings, is permutation-equivariant, the exact patch order has been treated as inconsequential.

However, this assumption does not hold in the context of modern, long-sequence models which introduce strong inductive biases such as locality [1], recurrence [2], or input-dependent state dynamics [3] that are sensitive to input ordering. Through mechanisms such as sparse attention via masking summarizing early parts of a long-sequence into latent representations, these methods are able to model long-sequences in a computationally tractable fashion. However, these design choices introduce a strong dependency on patch ordering, something that has previously been overlooked.

In this work, we demonstrate that merely swapping the row-major scan for an alternative, such as column-major or Hilbert curves, yields measurable accuracy gains across multiple long-sequence backbones. Further, we introduce *REOrder*, a method to learn an optimal patch ordering. *REOrder* first approximates which patch ordering may lead to best performance and then learns to rank patches in order of importance with reinforcement learning. Specifically, to convert the 2-D image into a 1-D sequence, we first quantify the compressibility resulting from each of six different patch orderings. Then, a Plackett-Luce ranking model is initialized with the least compressible ordering and further trained to minimize classification loss with REINFORCE. *REOrder* improves performance on ImageNet-1K by up to $3.01\%$ ($\pm0.23\%$) and Functional Map of the World by $13.35\%$ ($\pm0.21\%$). Code and animations are available on the project page.

39th Conference on Neural Information Processing Systems (NeurIPS 2025).

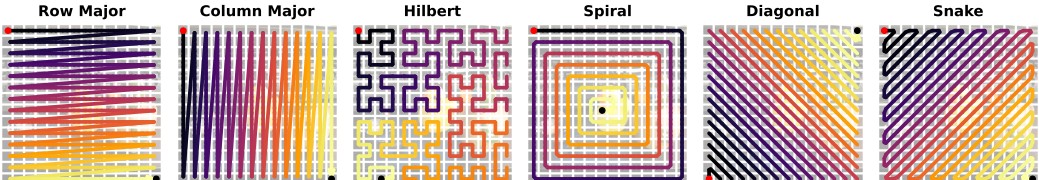

Figure 1: **Visualizations of alternate patch sequence orderings.** Six different patch orders—row-major, column-major, Hilbert curve, spiral, diagonal, and snake—are shown as trajectories over a $14 \times 14$ grid of patches. Each trajectory begins at the red dot and progresses to the black dot, illustrating the 1-D ordering imposed on the 2-D patch grid.

## 2 Related Works

**Transformers as sequence models for vision.** The concept of treating visual data as linearized sequences for generative modeling and other tasks has been explored for some time, with early work using models like LSTMs on sequences of visual frames [4]. Building on the success of transformers in image-sequence modeling, the Image Transformer [5] adapted this architecture to single images by dividing each image into patches and processing these patches as a 1-D sequence. To manage the computational cost for images, the original Image Transformer employed self-attention mechanisms constrained to local neighborhoods of patches. The Vision Transformer (ViT) [6] expanded this attention approach, applying global self-attention over the entire sequence of image patches. However, this global attention mechanism, while effective, is an $\mathcal{O}(n^2)$ operation in computation and space w.r.t. the number of patches, becoming computationally prohibitive for long-sequences. To overcome this challenge and enable the application of transformers to larger images, subsequent research has developed various strategies, including hierarchical tokenization and more efficient attention mechanisms. Hierarchical tokenization, for instance, involves processing images at multiple scales to reduce the sequence length at higher levels. Gupta et al. [7] show how such hierarchical tokenization can be used for tractably modeling the resulting long-sequences. Motivated by this history of linearizing 2-D images into 1-D sequences for transformer processing and the methods developed to handle the resulting sequence length and computational complexity, in this work, we explore whether the specific order in which a 2-D image is converted to a 1-D sequence matters for model performance.

**Long-sequence vision models.** As established, the quadratic computational cost of full self-attention with respect to sequence length makes processing long-sequences of patches very expensive for standard Vision Transformers. To address this, significant research has focused on developing efficient transformer architectures and long-sequence models that reduce this complexity, such as Sparse Attention [8], Longformer [1], Transformer-XL [2], and Mamba [3, 9]. While making modeling tractable, these methods often introduce specific inductive biases in how they process sequences, potentially making them sensitive to the order of the input tokens (which we discuss further in Section 3.3). In this work, we study the effect of patch ordering on Longformer, Transformer-XL, and Mamba, demonstrating that the inductive biases inherent in these different long-sequence modeling approaches lead to substantial accuracy variance across different patch orders.

**Patch order sensitivity.** Qin et al. [10] provided an in-depth study of the self-attention mechanism, establishing its property of permutation equivariance. However, a comparable analysis for long-sequence models that process vision tokens remains largely unexplored. This gap is significant, especially when considering findings from NLP, where studies like [11] revealed that long context models tend to neglect tokens in the middle of a sequence. Motivated by these observations and the underexplored nature of token ordering in long visual sequences, our work aims to investigate these effects. While some initial work in vision, such as ARM [9], has touched upon scan order for Mamba models (settling on small, row-wise clusters), this exploration did not comprehensively cover the wider space of possible orderings. Our research addresses this limitation by more rigorously examining the search space through experiments with six different orderings across multiple sequence models. Furthermore, to move beyond predefined arrangements, our method, *REOrder*, employs reinforcement learning to explore the space of possible permutations for the patch sequence.

**Learning to rank with reinforcement learning.** The foundation of learning to rank from pairwise comparisons was established by Bradley and Terry [12] who proposed a probabilistic model to estimate item scores via maximum likelihood. More recent work has extended learning to rank with reinforcement learning frameworks, particularly in information retrieval [13] and recommendation systems [14]. Wu et. al. [15] assign importance scores to image tokens based on their impact on CLIP's predictions then trains a supervised predictor to replicate these scores for efficient token pruning. Learning to rank patches can be reformulated to learning a permutation resulting from the ordering of patches by rank. Büchler et. al. learns to select effective permutations of patches or frames through a reinforcement policy that maximizes the improvement in self-supervised permutation classification accuracy in a discrete action space. In contrast, *REOrder* models a stochastic policy over permutations using a Plackett-Luce distribution [16, 17] and optimizes it with REINFORCE [18] and Gumbel Top-$k$ sampling which allows for more flexible orderings.

## 3 Preliminaries

We first establish the properties of the self-attention mechanism, specifically its equivariance under conjugation by permutation matrices and the matrix multiplication that lends it $\mathcal{O}(n^2)$ complexity. We then examine how Transformer-XL, Longformer, and Mamba model relationships between long-sequences and the design choices they make to side-step the quadratic complexity of self-attention. This establishes the sensitivity of long-sequence models to patch ordering.

### 3.1 Self-attention and permutation equivariance

Self-attention is the primary mechanism used in the Vision Transformer. Let the $n$ image patches form the matrix

$$\mathbf{X} = [x_1; \ldots; x_n] \in \mathbb{R}^{n \times d}.$$

Self-attention can then be computed as

$$\text{Attn}(\mathbf{X}) = \text{softmax}\left(\frac{(\mathbf{X}\mathbf{W}_q)(\mathbf{X}\mathbf{W}_k)^\top}{\sqrt{d}}\right) \mathbf{X}\mathbf{W}_v, \qquad \mathbf{W}_q, \mathbf{W}_k, \mathbf{W}_v \in \mathbb{R}^{d \times d} \tag{1}$$

where $\mathbf{W}_q$, $\mathbf{W}_k$, and $\mathbf{W}_v$ are learnable query, key, and value matrices, respectively.

The $n \times n$ similarity matrix inside the softmax is $\mathcal{O}(n^2)$ which is undesirable for large $X$. However, full self-attention has useful symmetry.

**Proposition 3.1** (Permutation equivariance of self-attention)**.** For every permutation matrix $\mathbf{P} \in \{0, 1\}^{n \times n}$,

$$\text{Attn}(\mathbf{P}\mathbf{X}) = \mathbf{P} \, \text{Attn}(\mathbf{X}). \tag{2}$$

Equation (2) states that self-attention is equivariant to arbitrary permutations of the patch ordering. Hence, the Vision Transformer, composed entirely of self-attention, is also permutation-equivariant. A proof for Proposition (3.1) is in Appendix B; a thorough analysis is conducted by Xu et. al. [19].

### 3.2 Position embeddings

The permutation equivariance of full self-attention is undesirable for image processing tasks. The spatial arrangement of patches in an image carries critical semantic information that must be preserved or made accessible to the model. Positional embeddings were introduced to explicitly provide the model with information about the original 2-D location of each patch within the image grid by summation with the image tokens. This allows the model to understand the spatial relationships between patches, regardless of their position in the input 1-D sequence.

We use learned absolute position embeddings across our models. In Transformer-XL, these are incorporated in conjunction with its native relative positional encoding. When a specific base patch ordering is adopted for an experiment, the positional embeddings are learned to align with that particular fixed sequence. Crucially, the positional embedding for the [CLS] token is consistently learned for the initial sequence position and is not reordered with the patch tokens, thereby maintaining a stable reference for classification tasks.

## 3.3 Self-attention approximations and sensitivity to patch order

Recent work reduces the $\mathcal{O}(n^2)$ cost of full attention by sparsifying the attention pattern, factoring the softmax kernel, or replacing it with a learned recurrence. These changes, however, also break the full permutation-equivariance guarantee of Eq. (2). Below we inspect three representative models. Their attention patterns are visualized in Appendix G.

**Transformer-XL.** Transformer-XL [2] adds segment-level recurrence with memory. After a segment of length $L$ is processed, each layer caches its hidden states as memory $\mathbf{M} \in \mathbb{R}^{m \times d}$. At the next step the layer concatenates that memory with the current segment's hidden states $\mathbf{H} \in \mathbb{R}^{L \times d}$:

$$\tilde{\mathbf{H}} = \text{concat}\big[\text{SG}(\mathbf{M}), \ \mathbf{H}\big] \in \mathbb{R}^{(m+L) \times d},$$

where $\text{SG}(\cdot)$ is a stop-gradient. Following Eq. (1), we define a single set of projection matrices as

$$Q = \mathbf{H}\mathbf{W}_q, \qquad K = \tilde{\mathbf{H}}\,\mathbf{W}_k, \qquad V = \tilde{\mathbf{H}}\,\mathbf{W}_v$$

Let $\mathbf{P}$ permute the $L$ patches of only the current segment (but not the memory). Memory corresponds to previous segments, so its rows are fixed under permutation.

$$\mathbf{P}\tilde{\mathbf{H}} = [\text{SG}(\mathbf{M}); \ \mathbf{P}\mathbf{H}]$$

Because the second and third logit terms of $\mathbf{M}_{\text{TXL}}(i, j)$ depend on the relative index $i - j$, conjugating by $\mathbf{P}$ does not commute with the soft-max:

$$A^{\text{TXL}}(\mathbf{P}\mathbf{H}) \neq \mathbf{P}\,A^{\text{TXL}}(\mathbf{H}).$$

Thus, Transformer-XL is permutation-sensitive even though its content-content term alone would be equivariant.

**Longformer.** Longformer [1] uses a sliding-window pattern of width $w$ plus $g$ global tokens with their own projections $\mathbf{W}_q^{(g)}, \mathbf{W}_k^{(g)}, \mathbf{W}_v^{(g)}$. Because the local mask $\mathbf{M}^{\text{local}}$ fixes which $(i, j)$ pairs are valid, applying $P$ to the rows/columns re-labels many entries as illegal ($-\infty$) and others as legal, so

$$\text{softmax}(\mathbf{P}\mathbf{M}^{\text{local}}\mathbf{P}^\top) \neq \mathbf{P}\,\text{softmax}(\mathbf{M}^{\text{local}})\,\mathbf{P}^\top.$$

The same holds for the global mask unless $\mathbf{P}$ preserves the hand-picked global positions. Therefore, Longformer is sensitive to patch ordering by design. It converts $\mathcal{O}(n^2)$ complexity to $\mathcal{O}(nw)$ under a rigid spatial prior.

**Mamba and ARM.** Mamba [3] does not implement quadratic attention and instead introduces a content-dependent state-space update.

$$\mathbf{h}_t = \mathbf{A}_t\,\mathbf{h}_{t-1} + \mathbf{B}_t\,\mathbf{x}_t, \qquad \mathbf{y}_t = \mathbf{C}_t\,\mathbf{h}_t,$$

with inputs $\mathbf{x}_t$ produced by scanning the patch sequence left-to-right. A permutation $\mathbf{P}$ re-orders the stream, changing $(\mathbf{B}_t, \mathbf{C}_t, \boldsymbol{\Delta}_t)$ and the sequence of matrix multiplications, so the recurrence yield is different. Therefore Mamba is also permutation sensitive. Its $\mathcal{O}(n)$ complexity comes at the price of a fixed processing order.

In this work, we use ARM as introduced by Ren et. al. [9]. In ARM, every Mamba layer runs four causal scans in different directions $d$ and sums their outputs before the channel-mixing MLP.

For each $d \in \{\rightarrow, \leftarrow, \downarrow, \uparrow\}$ let $x_t^{(d)}$ be the patch encountered at time step $t$ when traversing the image in direction $d$. The direction-specific recurrence is

$$\mathbf{h}_t^{(d)} = \mathbf{A}_t^{(d)}\,\mathbf{h}_{t-1}^{(d)} + \mathbf{B}_t^{(d)}\,\mathbf{x}_t^{(d)}, \qquad \mathbf{y}_t^{(d)} = \mathbf{C}_t^{(d)}\,\mathbf{h}_t^{(d)},$$

and the layer output combines the four scans via

$$\mathbf{y}_t = \sum_{d \in \{\rightarrow, \leftarrow, \downarrow, \uparrow\}} \mathbf{y}_t^{(d)}.$$

Similar to Mamba, because each scan is individually directional, an arbitrary permutation $P$ of the patch order both re-orders the input streams $\{\mathbf{x}_t^{(d)}\}$ and alters the learned parameter sequences $\{\mathbf{A}_t^{(d)}, \mathbf{B}_t^{(d)}, \mathbf{C}_t^{(d)}\}$. Hence the mapping still violates permutation equivariance $\mathbf{y}(\mathbf{P}\mathbf{X}) \neq \mathbf{P}\,\mathbf{y}(\mathbf{X})$. Further discussion in regards to the symmetry and performance implications of ARM with different orderings is discussed in Appendix A

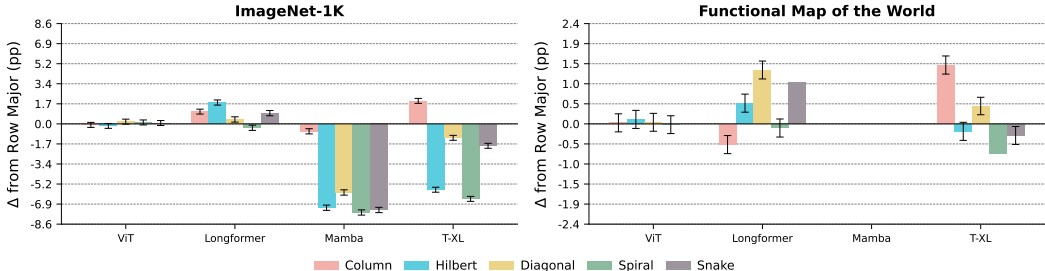

Figure 2: **Patch order affects the performance of long-sequence models.** This figure compares the top-1 accuracy of Vision Transformer (ViT), Longformer, Mamba, and Transformer-XL (T-XL) on ImageNet-1K and Functional Map of the World when using alternate patch orderings, relative to their standard row-major performance. As expected, ViT remains equivariant to patch sequence permutations. In contrast, long-sequence models exhibit substantial performance variability depending on the patch ordering. No single ordering consistently outperforms others across models or datasets, necessitating dynamic patch ordering strategies.

## 4 Does Patch Order Matter?

Conventional autoregressive vision models default to a raster-scan (row-major) order for flattening 2-D images into 1-D sequences. We investigate the question of whether the sequence in which image patches are presented to autoregressive vision models has an impact on their performance. We begin by outlining the datasets and models used in our empirical evaluation, followed by details of our training methodology. We then present results demonstrating that variations in patch ordering, even simple ones like column-major scans, can lead to differences in model outcomes, thereby motivating a search for more optimal, potentially learned, orderings. We define and explore six fixed patch orders: row-major, column-major, Hilbert curve, spiral, diagonal, and snake. These are visualized in Figure 1 and formally defined in Appendix F.

**Datasets.** Images captured in different contexts demonstrate varying structural biases. To study whether such datasets are susceptible to patch ordering effects to different degrees, we run experiments on two datasets: ImageNet-1K [20] (natural images) and Functional Map of the World [21] (satellite). We train on their respective training sets and report results on the validation sets. Details on the dataset licenses are provided in Appendix E

**Models.** We experiment with the Vision Transformer (ViT) [6], Transformer-XL (TXL) [2], Longformer [1], and ARM model as our Mamba variant [3, 9]. These three long-sequence architectures were chosen to represent distinct approaches to efficient sequence modeling: Transformer-XL employs segment-level recurrence and relative positional encodings to extend context length; Longformer reduces the quadratic attention cost through a combination of sliding-window and global attention; and Mamba introduces a structured state-space model with linear-time complexity and constant memory scaling. Together, they span the major design paradigms for long-range dependency modeling. We then test whether patch-ordering effects persist across fundamentally different inductive biases.

We utilized the `timm` implementation for the Vision Transformer (ViT) and the HuggingFace implementation for Longformer. Both were adapted with minor modifications to accommodate varying patch permutations. TXL is based on the official implementation and includes a newly introduced, learned absolute position embedding to account for changing patch orders across batches. We use ARM [9] as our vision Mamba model of choice due to its training stability. For all of the models, the image size is 224×224 with a patch size of 16×16. The Transformer-XL memory length ($\mathbf{M}$) was set to 128 and the attention window size ($\mathbf{M}^{local}$) for Longformer was set to 14. All four models prepend a learnable class `[CLS]` token as a fixed-length representation for image classification. The `[CLS]` token is always retained as the first token in the sequence. All models use their respective Base configurations. Complete details about the model configurations are in Appendix C.

**Training.** Experiments are conducted on machines equipped with either 8×80GB A100 GPUs or 4×40GB A100 GPUs. We apply basic data augmentations of resizing to 256×256, center crop to 224×224 and then a horizontal flip with $p = 0.5$. We ablate this augmentation choice in Appendix D.3. All models are trained for 100 epochs the AdamW optimizer using $\beta_1 = 0.9$,

$\beta_2 = 0.999$, weight decay of 0.03, and a base learning rate of $\alpha = 1.0 \times 10^{-4}$. Batch sizes are held constant for all runs across all model-dataset pairs (details in Appendix D). We apply cosine learning rate decay with a linear warmup over 5 epochs. For the reinforcement learning experiments introduced in Section 6, we use the same optimizer configuration but with a reduced base learning rate of $\alpha = 1.0 \times 10^{-5}$ and no decay.

## 4.1 Performance Variation Across Orderings

We evaluated top-1 accuracy on validation sets, estimating the Standard Error of the Mean (SEM) using a non-parametric bootstrap method with 2,000 resamples. Our analysis focused on how model performance varied with different patch orderings across datasets.

As anticipated, the Vision Transformer (ViT) demonstrated permutation equivariance, achieving consistent top-1 accuracy (approx. 37.5% on ImageNet-1K and 46.5% on FMoW) irrespective of patch order (Figure 2). In contrast, long-sequence models like Longformer, Mamba, and Transformer-XL (T-XL) showed performance variations dependent on patch order. T-XL and Mamba were particularly sensitive; on ImageNet-1K, T-XL's accuracy increased by 1.92%(±0.21 percentage points) with column-major ordering and decreased by 6.43%(±0.21 percentage points) with spiral. Longformer also benefited from alternative orderings (column-major, Hilbert, snake) on ImageNet-1K, improving by up to 1.83%(±0.22 percentage points) over row-major.

Dataset characteristics altered these trends. On FMoW, Longformer's optimal ordering shifted (e.g., diagonal increased accuracy by 1.3%(±0.22 percentage points) while column-major was detrimental). T-XL still favored column-major but also benefited from diagonal ordering on FMoW. Overall, FMoW exhibited less sensitivity to patch ordering than ImageNet, likely due to the greater homogeneity of satellite imagery compared to diverse natural images. Mamba consistently performed worse with non-row/column major orderings, likely because its fixed causal scan directions ($d \in \{\rightarrow, \leftarrow, \downarrow, \uparrow\}$) conflict with other patch sequences. Adapting Mamba's scan order to the patch ordering could potentially mitigate this performance drop.

These results highlight that there is no one ordering that works best for all models or datasets. Given that in many cases an alternate ordering outperforms the standard row-major order, this raises an important question: can we discover an optimal ordering tailored to a specific model and dataset? Moreover, are there useful priors we can identify to guide the search for such orderings?

## 5 Learning an Optimal Patch Ordering with *REOrder*

The observation that patch order influences model performance suggests the existence of an optimal ordering for each model-dataset pair. To find such orderings, we introduce *REOrder*, a unified framework that combines unsupervised prior discovery with task-specific learning. We begin with an information-theoretic analysis, examining the link between sequence compressibility and downstream performance. *REOrder* automates this process to derive a prior over effective patch orderings. Building on this, it employs a reinforcement learning approach to directly learn task-specific orderings, leveraging the prior as guidance.

**Information-Theoretic Initialization** The order in which image patches are arranged affects the compressibility of the resulting sequence. For example, in a conventional raster-scan order, adjacent patches often contain similar content, making the sequence more compressible. This local redundancy might make the prediction task more trivial, as the model could focus on learning simple local correlations rather than capturing more complex, long-range dependencies. We first explore whether compression metrics could serve as a proxy for evaluating different patch orderings. Specifically, we discretize images using a VQ-VAE based model [22] and encode the resulting token sequence codes using both unigram and bigram tokenization. For each configuration, we measure the compression ratio achieved by LZMA, which provides a quantitative measure of local redundancy in the sequence.

In Figure 3, we observe that different patch orderings indeed lead to varying compression ratios. The row-major order, which is commonly used in vision transformers, achieves higher compression ratios, suggesting strong local redundancy. Interestingly, the Hilbert curve ordering, which aims to preserve spatial locality, shows similar compression characteristics to row-major order. In contrast, the column-major/spiral orderings exhibit lower compression ratios, indicating less local redundancy.

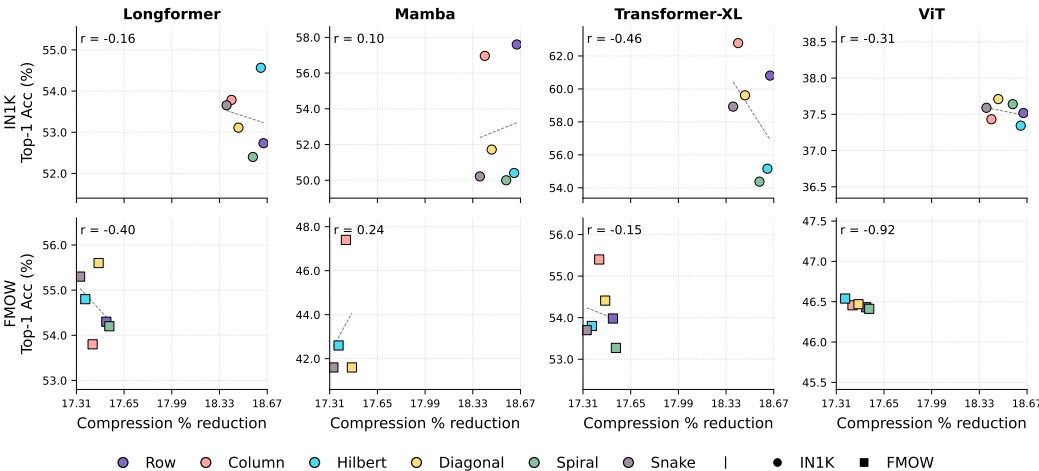

Figure 3: **Compression of 1-D sequences can serve as a weak prior for optimal patch ordering.** Top-1 accuracy is compared to percentage reduction for different patch orderings across four models for both ImageNet-1K and FMoW.

This result highlights the limitations of applying a 1D compression algorithm (LZMA) to sequences derived from 2D data. While Hilbert curves preserve 2D locality better, this may not translate to higher compressibility for a 1D algorithm.

While this analysis provides a useful lens for examining patch dependencies, it produces only a weak and model-dependent correlation with downstream accuracy. That is, lower compressibility does not consistently imply improved task performance. For instance, while column-major ordering in Figure 3 shows lower compression ratios, it does not consistently lead to better model performance across all architectures. This result is itself significant as it demonstrates that no simple, universal heuristic exists for a 1-D sequence of image tokens.

This insight motivates the second stage of our approach, in which we employ reinforcement learning to discover task-specific orderings. We use compressibility as a weak prior to give a warm start to our exploration. We assess the importance of such initializations in Appendix D.2.

## 6 Learning to Order Patches

Our findings in Section 4 reveal that patch ordering can significantly affect the performance of long-sequence vision models. This suggests the potential value of discovering a data- and model-specific ordering to improve task performance. Unfortunately, learning a discrete permutation poses a unique challenge. For an image with $N$ patches, there are $N!$ possible orderings. For $N = 196$, there are more than $10^{365}$ options. Searching this combinatorial space exhaustively is infeasible. A naïve approach would require evaluating each permutation's classification loss on every training example, which is computationally intractable and incompatible with gradient-based learning. We instead treat the selection of patch orderings as a stochastic policy learning problem, where the policy outputs a distribution over permutations. This allows us to sample permutations during training and optimize the policy against the downstream classification loss with reinforcement learning.

### 6.1 Learning the Permutation Policy via REINFORCE

Because permutations are discrete and non-differentiable, we adopt the REINFORCE algorithm [18, 23], a score-function estimator, to search over the space of all possible permutations and optimize a permutation policy. *REOrder* is outlined in Algorithm 1. This formulation treats the permutation sampler as a stochastic policy, whose parameters are updated based on the classification reward.

REINFORCE is unbiased but can have high variance. To mitigate this, we subtract a running baseline $b_t$ from the reward, giving $A_t = r_t - b_t$, where $b_{t+1} = \beta\, b_t + (1-\beta)\, r_t$, with $\beta\, (= 0.99)$ controlling the baseline's momentum. The vision transformer's parameters $\theta$ depend only on the cross-entropy loss $\mathcal{L}_{\mathrm{CE}}$, while the policy model receives gradients from $\mathcal{L}_{\mathrm{CE}}$ and the REINFORCE loss.

**Algorithm 1** *REOrder* with a Plackett-Luce policy

---

**Require:** mini-batch $\{(x^{(b)}, y^{(b)})\}_{b=1}^{B}$, backbone $f_\theta$, logits $z$, Gumbel temperature $\tau$, baseline $b$, momentum $\beta$
1: $g \sim \text{Gumbel}(0, 1)^n$
2: $\pi \leftarrow \text{argsort}_{\text{desc}}(z + \tau g)$                                  ▷ Gumbel-top-$k$ permutation
3: $x_\pi^{(b)} \leftarrow \text{PERMUTE}(x^{(b)}, \pi) \quad \forall b$
4: $\hat{y}^{(b)} \leftarrow f_\theta(x_\pi^{(b)})$
5: $\mathcal{L}_{\text{CE}} \leftarrow -\dfrac{1}{B} \sum_{b=1}^{B} \log \hat{y}_{y^{(b)}}^{(b)}$
6: $r \leftarrow -\mathcal{L}_{\text{CE}}$                                            ▷ Reward
7: $b \leftarrow \beta b + (1 - \beta) r$
8: $A \leftarrow r - b$                                              ▷ Advantage
9: $\mathcal{L}_{\text{policy}} \leftarrow -A \log P(\pi \mid z)$
10: $\mathcal{L}_{\text{total}} \leftarrow \mathcal{L}_{\text{CE}} + \mathcal{L}_{\text{policy}}$
11: Back-propagate $\nabla_{\theta, z} \mathcal{L}_{\text{total}}$ and update with Adam

---

### 6.2 The Plackett-Luce Policy for Patch Orderings

To parameterize the permutation distribution, we use the Plackett-Luce (PL) model [16, 17]. This model defines a distribution over permutations based on a learned logit vector $z \in \mathbb{R}^n$ associated with the image patches. Each patch gets a single parameter, so a 224×224 image with patch size 16×16 results in a model with 196 parameters. This is negligible added overhead for training. A permutation is sampled sequentially: at each step, a yet-unplaced patch is selected according to a softmax over the remaining logits:

$$P(\pi \mid z) = \prod_{i=1}^{n} \frac{\exp(z_{\pi_i})}{\sum_{k=i}^{n} \exp(z_{\pi_k})}. \tag{3}$$

Sampling from this distribution naively requires drawing one patch at a time in a sequential loop, which is inherently slow and difficult to parallelize across a batch. To make training efficient, we use the Gumbel-top-$k$ trick [24] which generates samples from the Plackett-Luce distribution by perturbing logits with Gumbel noise and sorting. A policy $\pi$ is sampled as $\pi = \text{argsort}(z + \tau g)$ where $g_i \sim \text{Gumbel}(0, 1)$ and $\tau > 0$ is a temperature parameter that trades off exploration and exploitation. The Gumbel-top-$k$ sampler is fully parallelizable and faster than iterative sampling, allowing permutation sampling in $\mathcal{O}(n \log n)$ time.

The log-probability of a sampled permutation is computed in closed form:

$$\log P(\pi \mid z) = \sum_{i=1}^{n} \Big[ z_{\pi_i} - \log \sum_{k=i}^{n} \exp(z_{\pi_k}) \Big], \tag{4}$$

which we implement efficiently using cumulative `logsumexp` in reverse.

The logit vector $z$ is initialized as a linear ramp from 0 to $-1$, then permuted according to the information-theoretic prior described in Section 5. This gives the model a sensible starting point that reflects structural cues discovered during unsupervised analysis, while still allowing gradients to adapt the ordering throughout training.

During training, a single set of Gumbel samples is drawn per batch so that every image shares the same permutation. At test time, we compute a deterministic maximum-likelihood permutation $\hat{\pi} = \text{argsort}(z)$. To ensure stability, we permute only the positional embeddings of image tokens with $\pi$, keeping the `[CLS]` token fixed.

### 6.3 Curriculum for Policy Learning

We adopt a three-stage curriculum that cleanly separates representation learning from policy learning. For the first $N$ epochs the classifier is trained with the canonical row-major patch order. This gives the vision transformer a stable starting point before any permutation noise is introduced. Beginning at epoch $N$ the Plackett-Luce policy is activated and trained with REINFORCE for $M$ epochs. Sampling uses the Gumbel-top-$k$ trick with a temperature schedule $\tau_t$ that climbs to a peak value

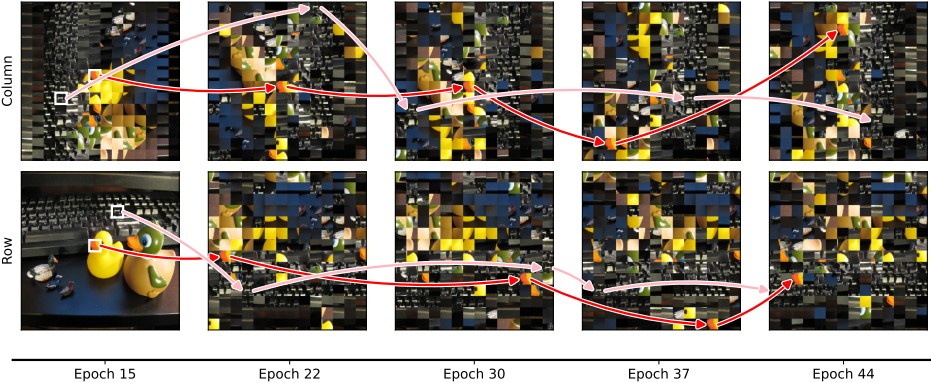

Figure 4: **The logits of the Plackett-Luce model, and therefore the permutation order, changes over the course of training**. Longformer is initialized with column and row-major patch ordering and optimized with *REOrder* on ImageNet-1K. The image is of the class "keyboard." We track two patches over the course of the policy curriculum: a keyboard key (light red arrow) and an irrelevant orange beak (dark red arrow). As the policy learns to order patches, we see the patches move toward the back of the sequence (i.e., are back-loaded) reflecting the dataset's center bias.

before decaying to zero. While $\tau_t > 0$ the permutation remains stochastic, encouraging exploration and allowing the policy to discover beneficial orderings. Once the temperature hits zero at epoch $N + M$, sampling collapses to the single maximum-likelihood permutation $\hat{\pi} = \text{argsort}(z)$. The policy gradients vanish, freezing the permutation, and the remaining epochs let the backbone finish optimizing with its now-determinate input order.

For our experiments. the curriculum was set with $N = 15$ and $M = 30$. $\tau_t$ follows a triangle pattern where it increases linearly from $\tau = 0.0$ at epoch $N$ to $M/2$ with $\tau = 0.2$. It then decreases linearly from epoch $M/2$ to $M$ ending at $\tau = 0$. The optimizer for the policy was AdamW($\beta_1$=0.9, $\beta_2$=0.999, weight decay 0.03) with a learning rate of $1 \times 10^{-4}$. The evolution of the policy over training is visualized in Figure 4, demonstrating how salient patches for the target class of "keyboard" move to the end of the sequence to maximize classification accuracy.

## 6.4 Effectiveness of Learned Patch Ordering

The Plackett-Luce policy introduced by *REOrder* is a simple drop-in addition to model training, and as visualized in Figure 5, in almost all cases, improves performance over their respective base patch order runs. Mamba observes gains of 2.20% ($\pm 0.22$ percentage points) on average across different patch orderings for IN1K, and 9.32% ($\pm 0.22$ percentage points) on FMoW. For ImageNet, the Hilbert curve ordering for Mamba is improved by 3.01% ($\pm 0.23$ percentage points) while for Functional Map of the World, the diagonal ordering is improved by 13.35% ($\pm 0.21$ percentage points). Transformer-XL demonstrates modest gains with *REOrder*. On IN1K, Transformer-XL accuracy improves by an average of 0.70% ($\pm 0.22$ percentage points) but sees significant gains with the Hilbert curve (1.50% ($\pm 0.21$ percentage points)) and spiral (1.09% ($\pm 0.21$ percentage points)) patch orderings. On FMoW, *REOrder* improves the best-performing column-major order by an additional 1.10% ($\pm 0.21$ percentage points), demonstrating that simply using a basic patch ordering alone may not be sufficient to get best performance. Longformer is unable to improve its accuracy on either dataset. However, since Longformer was the model that was initially also the least susceptible to changes in patch ordering due to its near-full approximation of self-attention, it is unsurprising that the use of *REOrder* does not achieve any noticeable performance gains.

To verify that these gains are not artifacts of training dynamics or limited training duration, we conducted two complementary robustness analyses (see Appendix D.2 and D.1). The first examines the role of policy-driven exploration by comparing static and dynamic random permutation strategies. The second extends Transformer-XL training from 100 to 300 epochs to test stability under longer optimization schedules. Together, these experiments confirm that performance improvements arise from the reinforcement learning process itself and persist under extended training.

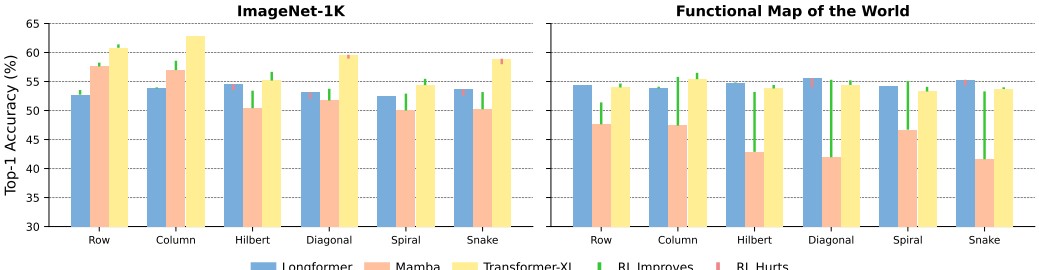

Figure 5: ***REOrder* finds improvements over the best patch ordering prior in almost all cases.** Across all models, *REOrder* can find a better patch ordering than a static prior and improve accuracy across both ImageNet-1K and Functional Map of the World.

In summary, *REOrder*'s approach of learning a patch ordering proves to be a broadly effective strategy for boosting classification performance with long-sequence models, consistently elevating results even beyond the strongest baseline patch orderings for nearly all tested models and datasets. We outline current limitations of our study and potential future directions in Appendix A. We also provide a thorough interpretability analysis in Appendix H.

## 7 Conclusion

This work establishes that contemporary vision models are sensitive to patch order and that row-major patch ordering, an extremely common way of converting 2-D images to 1-D sequences, can be suboptimal in many cases. Architectural modifications in models like Transformer-XL, Mamba, and Longformer, while enabling the processing of long-sequences, break permutation equivariance, leading to significant accuracy variations with different scan orders. We introduce *REOrder*, a method to learn optimal patch orderings by first deriving an information-theoretic prior and then learning a patch ranking by optimizing a Plackett-Luce policy with REINFORCE. This learned policy, implemented as a simple drop-in addition to model training, demonstrably improves classification accuracy for multiple long-sequence models on datasets such as ImageNet-1 and Functional Map of the World by up to by up to $3.01\%$ and $13.35\%$, respectively, surpassing the best fixed orderings.

## Acknowledgements

Thank you to Jiaxin Ge and Lisa Dunlap for generously reviewing this paper for clarity and content. Stephanie Fu lent her keen design instinct and helped make the figures look pretty. Anand Siththaranjan and Sanjeev Raja were always game for late night discussions about how to search over a combinatorial space and the pitfalls of reinforcement learning. As part of their affiliation with UC Berkeley, the authors were supported in part by the National Science Foundation, the U.S. Department of Defense, and/or the Berkeley Artificial Intelligence Research (BAIR) Industrial Alliance program. The views, opinions, and/or findings expressed are those of the authors and should not be interpreted as representing the official views or policies of any supporting entity, including the Department of Defense or the U.S. Government. This work utilized the infrastructure at the DoD's High Performance Computing Modernization Program (HPCMP).

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

# Appendix

## A   Discussion and Limitations

There are many details and nuances to our experiments that are worth discussing further. Particularly, there are areas for improvements that we would like to introduce in future versions of this work.

**Directional Asymmetry and ARM Interpretation.**   While our work primarily focuses on improving the permutation sensitivity of sequence-based vision models through *REOrder*, it is important to contextualize the architectural behavior of ARM. Although ARM's four-directional scanning mechanism is symmetric in principle, our experiments show that its performance varies across different input orderings due to the anisotropic statistics of natural images and the model's fixed directional composition. In particular, ARM achieves higher accuracy with a row-major ordering than with a column-major one, reflecting the stronger horizontal correlations inherent in natural scenes. This empirical asymmetry persists despite theoretical symmetry between the two configurations, revealing how data bias and architectural inductive bias jointly shape model behavior. Moreover, a Hilbert-curve ordering, which maximizes spatial locality, performs significantly worse because it disrupts ARM's directional continuity. ARM's recurrence dynamics are optimized for specific scanning patterns rather than spatial adjacency alone.

*REOrder* addresses these mismatches by learning an adaptive permutation of the input sequence that jointly benefits all four directional scans. When initialized from a row-major order, it improves validation top-1 accuracy to $58.25\%$, and from a column-major order to $58.58\%$ on ImageNet-1K. Even from a suboptimal Hilbert initialization, *REOrder* recovers much of the lost performance, reaching $53.42\%$. These results indicate that the learned permutations do not merely refine an existing scan but instead adaptively reorganize the patch sequence to better align the input statistics with ARM's directional processing. This demonstrates that *REOrder* can serve as a practical mechanism for reconciling architectural symmetry with anisotropic visual data.

**Random baseline.**   A random baseline for a learned permutation baseline involves sampling a random ordering of patches for every batch during training. We set up this baseline with Transformer-XL on the ImageNet-1K dataset. This model achieved a maximum top-1 accuracy of $39.07\%$ which is $15.25$ percentage points worse than the worst patch ordering tested for T-XL on IN1K (spiral). While a random baseline should be run for every model and dataset combination, the drastically lower performance of the random baseline gives us confidence in the veracity of our results.

**Under-explored and under-tuned policy.**   Images with $N = 196$ patches have more than $10^{365}$ permutations. Searching this space exhaustively is computationally infeasible (it is estimated that there are between $10^{78}$-$10^{82}$ atoms in the observable universe—our quantity is slightly larger than that). The Plackett-Luce policy exploration runs for a vanishingly small amount of time. The curriculum, as tested in our experiments, is only active for 30 epochs. The majority of these epochs is spent warming up and cooling down the Gumbel noise temperature, with the peak noise (and therefore exploration) occurring for only one epoch. We were not able to tune these parameters due to computational constraints. Therefore, much is left "on the table" with respect to improving results with *REOrder*.

**Dynamic Image Policy.**   A key limitation of the present work is that *REOrder* learns a single global ordering shared across all samples within a dataset. While this design enables clean isolation of ordering effects and ensures stable optimization, it constrains the model's flexibility in adapting to image-specific structures. A promising future direction is to extend *REOrder* toward dynamic, per-image ordering, where a lightweight policy network predicts ordering logits conditioned on patch embeddings. Such an approach would allow the ordering to adapt to spatial and semantic content, potentially capturing intra-dataset variability while preserving the benefits of learned sequencing. Investigating the trade-offs between global stability and dynamic adaptability represents an important next step for our future work.

**Long Sequence Training.**   To assess scalability, we initiated experiments on ultra–high-resolution images (5888×5888), a setting where standard ViT models exceed GPU memory limits. These

experiments are ongoing, and we plan to include results in future work exploring how *REOrder* behaves in extended sequence regimes.

## B  Proof of Proposition 3.1

**Theorem 1** (Permutation equivariance of self-attention). *Let* $\mathbf{X} \in \mathbb{R}^{n \times d}$ *and let* $\mathbf{P} \in \{0, 1\}^{n \times n}$ *be any permutation matrix. With* $\mathrm{Attn}(\cdot)$ *defined in Eq. (1), we have*

$$\mathrm{Attn}(\mathbf{PX}) = \mathbf{P} \ \mathrm{Attn}(\mathbf{X}).$$

*Proof.* Let

$$S(\mathbf{X}) = \mathrm{softmax}\left(\frac{(\mathbf{XW}_q)(\mathbf{XW}_k)^\top}{\sqrt{d}}\right) \in \mathbb{R}^{n \times n}.$$

Conventionally, softmax for $\mathrm{Attn}$ is applied row-wise over $(\mathbf{XW}_q)(\mathbf{XW}_k)^\top$ so that the ensuing multiplication by the value matrix serves as a normalized weighted sum over value columns. Since a permutation merely re-orders rows and columns, it satisfies the conjugation property

$$\mathrm{softmax}(\mathbf{PMP}^\top) = \mathbf{P} \ \mathrm{softmax}(\mathbf{M}) \, \mathbf{P}^\top \tag{5}$$

for any square matrix $\mathbf{M}$ and any permutation matrix $\mathbf{M}$.

Then,

$$\mathrm{Attn}(\mathbf{PX}) = \mathrm{softmax}\left(\frac{(\mathbf{PXW}_q)(\mathbf{PXW}_k)^\top}{\sqrt{d}}\right) \mathbf{PXW}_v$$

$$= \mathrm{softmax}\left(\frac{\mathbf{P}(\mathbf{XW}_q)(\mathbf{XW}_k)^\top \mathbf{P}^\top}{\sqrt{d}}\right) \mathbf{PXW}_v$$

$$\overset{(5)}{=} \mathbf{P} \, S(\mathbf{X}) \, \mathbf{P}^\top \, \mathbf{PXW}_v$$

$$= \mathbf{P} \, S(\mathbf{X}) \, \mathbf{XW}_v \qquad\qquad (\mathbf{P}^\top \mathbf{P} = \mathbf{I})$$

$$\overset{(1)}{=} \mathbf{P} \ \mathrm{Attn}(\mathbf{X}).$$

Hence self-attention is permutation equivariant. □

## C  Model Details

Table 1: Patch Order Models

| Model | Size | # of Parameters | Width | Depth |
|---|---|---|---|---|
| ViT | Base | 86 570 728 | 768 | 12 |
| Transformer-XL | Base | 93 764 584 | 768 | 12 |
| Longformer | Base | 107 683 048 | 768 | 12 |
| Mamba | Base | 85 036 264 | 768 | 12 |
| ViT | Large | 304 330 216 | 1024 | 24 |
| Transformer-XL | Large | 329 601 512 | 1024 | 24 |
| Longformer | Large | 379 702 760 | 1024 | 24 |
| Mamba | Large | 297 041 352 | 1024 | 24 |

We attempt to parameter match each model we experiment with in an effort to remove one axis of variability from our results. We experiment with the Base variant of each model in our experiments. Despite the intention for the Base variants to be roughly equivalent to each other, Longformer-Base and Mamba-Base vary by ~22M parameters. To contextualize how width and depth affect the number of parameters, we provide details for Base and Large variants in Table 1.

# D Additional Model Training Details

Each training run required 100 epochs for completion. ImageNet experiments were run on $8\times$ 80GB A100 GPUs, while Functional Map of the World experiments were run on $4\times$ 40GB A100 GPUs. Transformer-XL, ViT and Mamba, and the Plackett-Luce policy were compiled with TorchDynamo using the Inductor backend. The Longformer encoder provided by HuggingFace was unable to be compiled. For ImageNet runs, Vision Transformer runs took ~8.5 hours to run, Transformer-XL runs took ~10 hours, Longformer runs took ~12 hours, and Mamba runs took ~19 hours. For Functional Map of the World, Vision Transformer runs took ~17.5 hours, Transformer-XL runs took ~21 hours, Longformer runs took ~26 hours, and Mamba runs took ~31 hours. All runs were executed on dedicated datacenters accessed remotely.

All runs nearly maximized the available VRAM on their respective GPUs. The breakdown of models and batch sizes is provided below:

Table 2: Batch sizes for every model-dataset pairing.

| Model | Size | Dataset | Batch Size |
|---|---|---|---|
| ViT | Base | IN1K | 896 |
| ViT | Base | FMOW | 448 |
| Transformer-XL | Base | IN1K | 640 |
| Transformer-XL | Base | FMOW | 320 |
| Longformer | Base | IN1K | 640 |
| Longformer | Base | FMOW | 320 |
| Mamba | Base | IN1K | 640 |
| Mamba | Base | FMOW | 320 |

In sum, the total set of experiments run required a total of $\sim 9,336$ A100 GPU hours.

Models were trained with a minimal set of augmentations, namely: resize to $256 \times 256$ pixels, center crop to $224 \times 224$ pixels, and a random horizontal flip with $p = 0.50$.

## D.1 Increased Training Time

We continued training our Transformer-XL Row and Transformer-XL *REOrder* Row models from 100 epochs to 300 epochs to test the robustness of our conclusions with longer training. The models were trained with the same recipe and learning curriculum. The gains from *REOrder* are maintained and even increased after 300 epochs. At 100 epochs (Table 3), Transformer-XL Row achieves $60.80\%$ Top-1 Validation accuracy whereas *REOrder* improves on this by $0.6$ percentage points to $61.40\%$. Extending training to 300 epochs, Transformer-XL Row achieves $61.20\%$, a $1.19$ percentage points increase over 100 epochs; however, Transformer-XL *REOrder* Row, increases performance to $63.34\%$ which is a $2.14$ percentage points increase over the Transformer-XL Row model at the same epoch. These findings show that *REOrder* continues to provide consistent gains under extended optimization, supporting its robustness beyond the initial training regime.

Table 3: ***REOrder* widens the gap between standard models when training is extended.** When training is extended from 100 epochs to 300 epochs, *REOrder* still shows significant Top-1 Validation Accuracy gains over training without *REOrder*.

| Model | Dataset | Patch Order | RL | Top-1 Validation Accuracy | Gain at 300 Epochs |
|---|---|---|---|---|---|
| T-XL | IN1K | Row | - | 60.80 | +1.19 *pp* |
| T-XL | IN1K | Row | ✓ | 61.40 | +1.94 *pp* |

## D.2 Random and Static Permutation Results

To better understand the effects of the *REOrder* learning process on the performance of the models we performed three additional runs with Transformer-XL on ImageNet-1k:

1. Using a fixed random permutation for the entire training

2. Sampling a random permutation for every batch through all of training

3. Using the final learn permutation of our best performing *REOrder* Transformer-XL model for all of training

The results in Table 4 indicate that simply picking random patch orderings is not sufficient (and therefore, any one random ordering will also perform poorly). The structured exploration provided by *REOrder* is necessary to improve performance. A novel outcome of this exploration is that simply using a learned patch ordering is not sufficient for gains in performance; the guided exploration process introduced by *REOrder* is necessary for improved performance.

Table 4: **Top-1 validation accuracy on ImageNet-1K using Transformer-XL under four alternative permutation schemes.** Each variant controls how patch sequences are presented during training. Only *REOrder*'s reinforcement-learning curriculum (second row) adaptively explores the ordering space, leading to a large performance gain over all static or unguided random strategies.

| Permutation Strategy | RL | Top-1 Accuracy |
|---|---|---|
| Row Major | - | 60.80 |
| *REOrder* Row-major initialization | ✓ | 61.40 |
| One random permutation for all epochs | - | 53.10 |
| Every batch random | - | 50.23 |
| Best learned ordering | - | 53.07 |

## D.3 Results Without Flips

We additionally run experiments where the only data processing steps applied are a resize to $256 \times 256$ pixels and a center crop to $224 \times 224$ pixels with no flips at all. The results presented in Table 5 show the effects observed in the main experiments are maintained albeit with different accuracy magnitudes. With no augmentations, the effect of *REOrder* is even greater. All tested patch orders exhibit performance gains when optimized by *REOrder*. The performance boost ranges from $1.58\%$ with row-major to $2.94\%$ with the Hilbert curve ordering. Diagonal and spiral orderings get moderate gains of $2.41\%$ and $2.47\%$ respectively, while column-major and snake orderings show more modest improvements of $1.67\%$ and $1.82\%$ respectively. These results confirm that *REOrder* consistently enhance task accuracy across the patch orderings even without extensive data augmentation.

Table 5: **REOrder consistently improves performance across all patch orders with no augmentations.** Top-1 validation accuracy on IN1K using different patch orders with and without our *REOrder* using minimal data processing (resize and center crop only).

| Model | Dataset | Patch Order | RL | Top-1 Accuracy | Difference |
|---|---|---|---|---|---|
| Transformer-XL | IN1K | Row | | 57.95 | |
| Transformer-XL | IN1K | Row | ✓ | 59.53 | +1.58% |
| Transformer-XL | IN1K | Column | | 57.33 | |
| Transformer-XL | IN1K | Column | ✓ | 59.01 | +1.67% |
| Transformer-XL | IN1K | Diagonal | | 52.79 | |
| Transformer-XL | IN1K | Diagonal | ✓ | 55.20 | +2.41% |
| Transformer-XL | IN1K | Hilbert | | 50.29 | |
| Transformer-XL | IN1K | Hilbert | ✓ | 53.22 | +2.94% |
| Transformer-XL | IN1K | Spiral | | 49.93 | |
| Transformer-XL | IN1K | Spiral | ✓ | 52.40 | +2.47% |
| Transformer-XL | IN1K | Snake | | 52.20 | |
| Transformer-XL | IN1K | Snake | ✓ | 54.02 | +1.82% |

# E  Dataset Licenses and References

In this work, we use ImageNet-1K as provided by the 2012 ImageNet Large-Scale Visual Recognition Challenge [20] and the RGB version of Functional Map of the World [21]. ImageNet-1K is obtained from the official download portal and Functional Map of the World is obtained from their official AWS S3 bucket.

ImageNet-1K is licensed non-commercial research and educational purposes only as described on their homepage. Functional Map of the World is licensed under a custom version of the Creative Commons license available on their GitHub.

# F  Rasterization Orders

Let an image be composed of $N = H \times W$ patches, arranged in a grid of height $H$ (number of rows) and width $W$ (number of columns). Each patch is identified by its zero-indexed 2D coordinates $(r, c)$, where $0 \leq r < H$ and $0 \leq c < W$. A rasterization order (or scan order) $\pi$ is a bijection that maps a 1D sequence index $k \in \{0, 1, \ldots, N - 1\}$ to the 2D coordinates of the $k$-th patch in the sequence. We denote this mapping as $\pi(k) = (r_k, c_k)$. In all cases besides row- and column-major, a direct formula for $(r_k, c_k)$ from $k$ is not provided; the order is algorithmically defined by their respective procedures.

## F.1  Row-Major Order

Row-major order, also known as raster scan, is the most common default. Patches are scanned row by row from top to bottom. Within each row, patches are scanned from left to right. The coordinates $(r_k, c_k)$ for the $k$-th patch are given by:

$$r_k = \left\lfloor \frac{k}{W} \right\rfloor$$
$$c_k = k \pmod{W}$$

## F.2  Column-Major Order

In column-major order, patches are scanned column by column from left to right. Within each column, patches are scanned from top to bottom. The coordinates $(r_k, c_k)$ for the $k$-th patch are given by:

$$c_k = \left\lfloor \frac{k}{H} \right\rfloor$$
$$r_k = k \pmod{H}$$

## F.3  Hilbert Curve Order

The Hilbert curve is a continuous fractal space-filling curve. Ordering patches according to a Hilbert curve aims to preserve locality, meaning that patches close in the 1D sequence are often (but not always perfectly) close in the 2D grid. The coordinates $(r_k, c_k)$ for the $k$-th patch are given by:
$$\pi(k) = (r_k, c_k) = \mathcal{H}_{H,W}(k)$$
where $\mathcal{H}_{H,W}(k)$ denotes the coordinates of the $k$-th point generated by a Hilbert curve algorithm adapted for an $H \times W$ grid.

## F.4  Spiral Order

In a spiral scan, patches are ordered in an outward spiral path, typically starting from a corner patch, e.g., $(0, 0)$. The path moves along the perimeter of increasingly larger (or remaining) rectangular sections of the grid. The coordinates $(r_k, c_k)$ for the $k$-th patch are:
$$\pi(k) = (r_k, c_k)$$
These are the $k$-th unique coordinates visited by a path that starts at $(0, 0)$ and spirals outwards. The path typically moves along segments of decreasing lengths (or until a boundary or previously visited cell is encountered) in a sequence of cardinal directions (e.g., right, down, left, up, then right again with a shorter segment, etc.), effectively tracing successive perimeters of nested rectangles.

### F.5 Diagonal Order (Anti-diagonal Scan)

In a diagonal scan, patches are ordered along anti-diagonals, which are lines where the sum of the row and column indices $(r + c)$ is constant. These anti-diagonals are typically scanned in increasing order of this sum, starting from $r + c = 0$. Within each anti-diagonal, patches are typically ordered by increasing row index $r$ (or, alternatively, by increasing column index $c$). The coordinates $(r_k, c_k)$ for the $k$-th patch are:

$$\pi(k) = (r_k, c_k)$$

such that $(r_k, c_k)$ is the $k$-th patch when all patches $(r, c)$ are ordered lexicographically according to the tuple $(s, r')$, where $s = r + c$ is the anti-diagonal index and $r' = r$ is the row index. Patches with smaller anti-diagonal sums $s$ come first. For patches on the same anti-diagonal (i.e., with the same $s$), those with a smaller row index $r'$ come first.

### F.6 Snake Order

This scan traverses patches along anti-diagonals (where the sum of row and column indices, $s = r + c$, is constant). The anti-diagonals are processed in increasing order of $s$. The direction of traversal along each anti-diagonal alternates. For example, for even $s$, patches are visited by increasing column index $c$, and for odd $s$, by decreasing column index $c$.

Let $s_k = r_k + c_k$ be the anti-diagonal index for the $k$-th patch $\pi(k) = (r_k, c_k)$. The sequence of patches $\pi(k)$ is generated by iterating $s$ from 0 to $H + W - 2$. For each $s$:

1. Define the set of coordinates on the anti-diagonal: $S_s = \{(r, c) \mid r + c = s, 0 \leq r < H, 0 \leq c < W\}$.

2. Order the coordinates in $S_s$. For instance, by increasing column index $c$: $P_s = [(r_0, c_0), (r_1, c_1), \ldots, (r_m, c_m)]$ where $c_0 < c_1 < \cdots < c_m$.

3. If $s$ is odd (or based on an alternating flag), reverse the order of $P_s$.

4. Append the (potentially reversed) $P_s$ to the overall sequence.

## G Attention Patterns for Tested Models

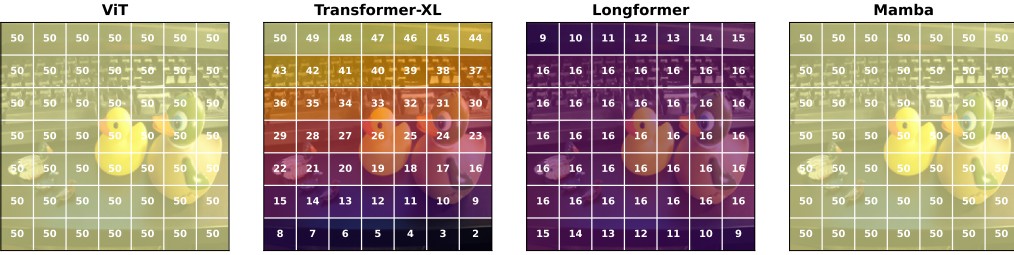

Figure 6: **Token-level attention coverage across different model architectures.** Each grid cell represents a patch in a 224×224 input image split into 49 non-overlapping 32×32 patches, plus a `[CLS]` token. Numbers indicate how many tokens attend to each patch. ViT and Mamba exhibit full attention to all patches. In contrast, Transformer-XL's causal attention and Longformer's local attention restrict the number of tokens that can attend to each patch, leading to a strong asymmetry and localized attention, respectively.

To produce Figure 6, we visualized how many tokens attend to each patch in a 224×224 image divided into 49 patches (plus a `[CLS]` token), using the following methods for each model.

**ViT and Transformer-XL:** We extracted the raw attention weights across all layers and heads during a forward pass. After computing the element-wise maximum across layers and heads, we obtained a binary $N \times N$ matrix indicating whether token $i$ attends to token $j$. Summing over rows yields how many tokens attend to each patch.

**Longformer:** Due to its local attention structure, we reconstructed a dense $N \times N$ attention matrix by extracting local and (if applicable) global attention indices. We then counted how many tokens had access to each patch through these sparse connections.

**Mamba:** Since Mamba does not use attention, we used a gradient-based saliency method. We computed the gradient of the $L_2$ norm of each output token with respect to the input embeddings. This yielded a sensitivity matrix indicating the influence of each input token on each output. Thresholding non-zero entries allows us to analogously count how many tokens "attend" to each patch.

These results reveal how the structural design of each model affects its ability to aggregate spatial information. ViT and Mamba attend to all patches uniformly, while Transformer-XL's causal structure and Longformer's locality lead to uneven and limited attention coverage. These patterns explain their respective sensitivities to input token ordering.

## H Policy Evolution During Training

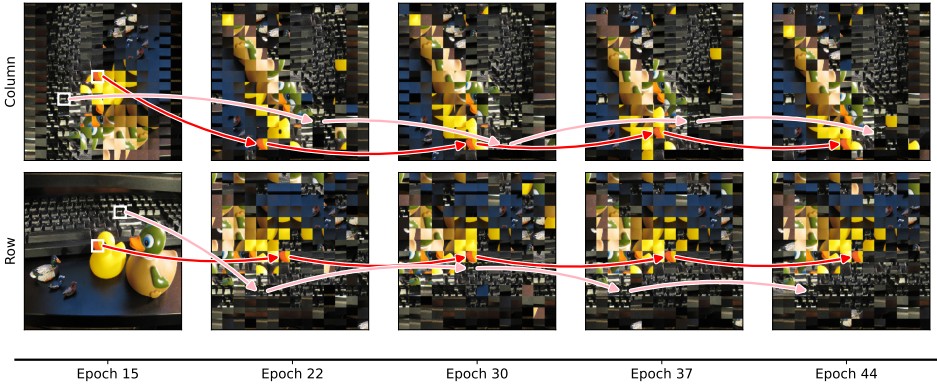

Figure 7: **Mamba observes patches most related to the class label move to the end of the sequence.** Mamba is trained with column- (top) and row-major (bottom) patch orderings and optimized with *REOrder*. The image is of the class "keyboard." We track two patches over the course of the policy curriculum: a keyboard key (light red arrow) and an irrelevant orange beak (dark red arrow). As training progresses, the keyboard-related patches shift into the final indices of the sequence.

The *REOrder* policy learned distinct and consistent rearrangements of image patches that reflect both dataset structure and model-specific inductive biases. As seen in the "keyboard" example from

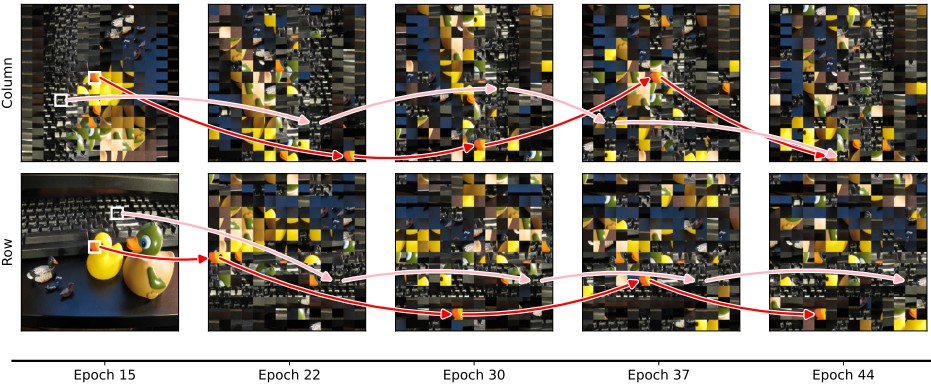

Figure 8: **Transformer-XL observes patches most related to the class label move to the end of the sequence.** Transformer-XL is initialized with column- (top) and row-major (bottom) patch orderings and optimized with *REOrder*. The image is of the class "keyboard." We track two patches over the course of the policy curriculum: a keyboard key (light red arrow) and an irrelevant orange beak (dark red arrow). As training progresses, the keyboard-related patches shift into the final indices of the sequence.

Figure 5, *REOrder* progressively shifts semantically important patches toward specific sequence positions, indicating that the learned permutations encode spatial continuity and capture architectural preferences in how visual information should be processed.

To quantify these effects, we analyzed how *REOrder* repositions patches originating from the central region of an image relative to their initial raster order. Both ImageNet-1K and FMoW exhibit a pronounced center bias, where salient content typically lies near the image's center. We observed that *REOrder* with different long-sequence architectures exploit this bias in distinct ways.

Mamba consistently front-loads central patches, shifting them earlier in the sequence (mean shift of $+6.8$ positions). This ordering aligns with Mamba's recurrent state-space mechanism as processing salient central regions early allows the model to form a strong initial hidden state that informs subsequent token processing.

Transformer-XL and Longformer exhibit the opposite tendency, back-loading central patches (average shifts of $-2.2$ and $-20.4$ positions, respectively). For Transformer-XL, this allows these tokens to utilize information from the entire image rather than only from half of it. For Longformer, the attention window accumulates context over the sequence which positioning maximizes contextual support from earlier tokens for the salient tokens.

This contrast demonstrates that *REOrder* does not converge to a trivial or dataset-wide ordering. Instead, it learns model-dependent strategies that align spatial inductive biases with sequence-processing dynamics, revealing that ordering optimization can adapt meaningfully to the architecture's internal computation pattern.

