# OpenReview forum: "REOrdering Patches Improves Vision Models"
_NeurIPS.cc/2025/Conference — NeurIPS 2025 poster_

### Official Review · Reviewer_zkbe · 2025-06-29

**Clarity:** 4
**Significance:** 2
**Originality:** 3
**Rating:** 4
**Confidence:** 3

**Summary:**

Applying a sequence model on image patches is an increasingly common paradigm in modern computer vision. This paper explores the impact of patch ordering on performance when full self-attention is dropped in favor of more efficient (but permutation-sensitive) alternatives like Mamba, Longformer, and Transformer-XL. This paper finds that simple alternatives to the usual row-major patch order (such as column-major and hilbert order) yield significantly different image classification performance on ImageNet-1K and Functional Map of the World. However, the optimal ordering varies per dataset-model pair and is only weakly correlated with compressibility, so the paper proposes a novel RL formulation to learn the best patch ordering by optimizing a Plackett-Luce distribution over permutations with a REINFORCE objective.

**Questions:**

I’ve embedded most of my questions and suggestions for the authors within the Strengths and Weaknesses section.

One other miscellaneous suggestion: Consider training the random permutation baselines for each model-dataset pair and including them in Figure 2. The paper slates this for future work, but if done, it’ll better contextualize the performance of the 6 explicitly considered orderings.

**Ethical Concerns:**

["NO or VERY MINOR ethics concerns only"]

**Final Justification:**

The paper's strengths are its novel premise, RL formulation for learning patch ordering, and clarity in its explanation of the proposed method. Rebuttals have largely clarified some misunderstandings in the narrative of the paper, which were originally leading to unclear takeaways. Nevertheless, the primary weakness of the work remains in that evaluations are done in a regime where full self-attention is quite feasible. For this reason, I'm keeping my original score.

**Limitations:**

Yes.

**Quality:**

2

**Strengths And Weaknesses:**

**Strengths**

1. **Novel premise and RL formulation**. To the best of my knowledge, this is the first paper that explores patch ordering for vision models. As vision models are scaled, the proposed ideas may become increasingly relevant. The resulting RL formulation to optimize over a distribution of permutations is also elegant and can serve as a template, even for other use cases.
2. **Clarity**. The paper is very well-written and does a great job of prefacing its contributions with bits of essential background, such as on Mamba, Longformer, Transformer-XL, and the Pluckett-Luce distribution.

**Weaknesses**

1. **Unclear takeaways**. The paper does a great job of probing the impact of patch ordering and bringing to light that the typical row-major order may not always be optimal. However, it remains unclear what the actual optimal patch ordering may be. Some extra analysis may help pick out the signal from the noise here. Questions I have in this direction:
   - On lines 205-208, the paper notes that Mamba/ARM doesn’t perform well with many orderings, likely because of its fixed scan directions (forwards, backwards, up, down). Can you visualize these scan orders? I can imagine what the forwards and backwards scan orders may look like, but what about up and down? If those scan orders contain many spatial discontinuities, that may explain the poor results and render many of those results uninterpretable. If so, can you adapt the scan orders or the patch orderings appropriately, to induce better performance?
   - Compression percent is only weakly correlated with performance, and the sign of the correlation also seems to depend on the model. Can you do a study on the robustness or noisiness of this metric? Why is Hilbert ordering less compressible than column ordering? This is surprising, as the paper points out. Could it be a sign of a noisy measure of compressibility? Are there other metrics that may show stronger correlations, e.g. using JPEG compression?
   - What do the patch orderings that are learned through RL look like? Do they converge to any patterns, by model or by dataset? Similarly, in Figure 4, why should the patches related to the “keyboard” class move to the end of the patch order? Was that trained on a single image? If not, what’s special about the “keyboard” class such that it dictates the optimal patch order across the entire dataset?
2. **Missing evaluation in the long sequence regime**. For all experiments in the paper, resolution is set to 224x224 input images, a regime where O(n^2) full self-attention is viable. In this regime, patch ordering is not really a consideration, since full self-attention is permutation invariant. Results would be more persuasive if tested on a truly long sequence regime, such as higher resolution images or videos.

Addressing these weaknesses may improve my rating.

---

> ### Author Rebuttal · Authors · 2025-07-30
>
> We sincerely thank the reviewer for their detailed and insightful review. We are thrilled that you found our paper to have a novel premise, an elegant RL formulation, and to be well-written. The weaknesses you've identified are helpful and highlight areas where we can provide deeper analysis. We agree with your assessments and address each of your points below:
>
> **ARM scan ordering and spatial continuity**: ARM does not use a single scan but instead uses an ensemble of four fixed, competing scan directions and sum their outputs to form the final representation: (1) forward row-major (top-left to bottom-right), (2) backward row-major (bottom-right to top-left), (3) forward column-major (top-left to bottom-right), and (4) backward column-major (bottom-right to top-left). Figure 1 in our paper visualizes row-major and column-major scan orders.
>
> A standard row-major patch ordering is optimal for the row-forward scan, but it is simultaneously suboptimal for the row-backward scan and the two column-based scans. This creates an inherent architectural tension. The final output of the Mamba block depends on the sum of these four scans, so the ideal input sequence is not one that is perfect for a single scan, but one that provides the best compromise for all four simultaneously.
>
> REOrder, meanwhile, does not re-learn a basic row-major or column-major sequence. Instead, it learns a permutation of that base ordering. Therefore, we learn a data- and model-specific patch sequence to maximize the combined effectiveness of the entire multi-scan ensemble. This provides improved performance against even a row-major or column-major patch order, as seen in Figure 5.
>
> **The information-theoretic approach is a weak prior**: Our exploration of compressibility was intended to see if a simple, information-theoretic heuristic could serve as a useful prior. The resulting weak and model-dependent correlation is, in itself, a key finding of our paper: it demonstrates that no simple, universal prior across a search space spanning ~10^{365} options exists, thus motivating our more powerful (but complex) reinforcement learning approach. Your point about Hilbert vs. column ordering is excellent and likely highlights the limitations of applying a 1D compression algorithm (LZMA) to sequences derived from 2D data. While Hilbert curves preserve 2D locality better, this may not translate to higher compressibility for a 1D algorithm.
>
> JPEG compression is a good suggestion for an alternative compression strategy to explore. Unfortunately, JPEG is designed for continuous-tone 2D images. Since the sequence models we test operate on 1D sequences, JPEG is not an applicable compression scheme that we can test. In fact, JPEG on shuffled 1D sequences, such as those following a Hilbert curve, would likely result in sequences that are less compressible than row-major due to sharp patch boundaries.
>
> **Interpretability of learned orderings**: We thank the reviewer for this point, which prompted us to conduct a deeper analysis of the patterns learned by REOrder. The “keyboard” example in Figure 4 suggested a link to semantic importance, but we agree a broader analysis is needed. Our new investigation reveals that REOrder learns to align the spatial structure of a dataset with the specific inductive biases of different model architectures.
>
> Our analysis is grounded in two key observations: (1) long-sequence models are sensitive to input order, and (2) ImageNet-1K and Functional Map of the World (FMoW) exhibit a strong center bias, where the most semantically critical content is usually in the middle of the image. To investigate this, we quantified how REOrder repositions patches from the image’s central region relative to their initial ordering. This can reveal whether the learned policy prioritizes central patches by moving them to the beginning (“front-loading”) or the end (“back-loading”) of the sequence. We discovered two distinct, model-dependent strategies:
>
> 1. Mamba prefers to front-load central tokens. REOrder systematically moved central patches towards the start of the sequence (on average, a shift of +6.8 positions). This “front-loading” strategy intuitively aligns with Mamba's recurrent state-space mechanism. By processing the most salient patches first, the model establishes a strong initial state ($h_t$​) that informs the processing of the entire sequence. This allows the model to build a robust understanding of the image’s core subject from the outset.
>
> 2. Longformer and Transformer-XL prefer to back-load central tokens. Both models learned to shift central patches towards the __end__ of the sequence (average shifts of -20.4 and -2.2 positions, respectively). Transformer-XL’s recurrent memory can summarize information from earlier, less critical patches. By positioning central patches later in the sequence, these tokens can utilize information from the entire image rather than only from half of it. Comparatively, Longformer’s sliding-window attention, which accumulates context over time, would provide greater information for patches near the end. This could maximize the contextual information available for the local and global attention mechanisms when classifying objects.
>
> This contrast between “front-loading” for Mamba and “back-loading” for Transformer-XL/Longformer on the same ImageNet dataset demonstrates that REOrder is not finding a single, trivial ordering. Instead, it is learning sophisticated, model-specific policies that exploit how each architecture processes information. We will add this detailed analysis and discussion to the final version.

---

> > ### Comment · Reviewer_zkbe · 2025-08-03
> > **Some follow-up questions**
> >
> > Thank you for your response!
> >
> > Some follow-up questions:
> > 1. **ARM scan ordering and spatial continuity.**
> >    - To verify my understanding, since ARM combines the four scan directions, wouldn't row-major, column-major, row-backwards-major, and column-backwards-major yield theoretically identical results?
> >    - For the same reason, wouldn't a Hilbert base ordering yield lots of spatial discontinuities for the column-major and column-backwards-major orders? Is it even fair to consider such an order for ARM, given that a key motivation of the Hilbert ordering is to minimize spatial discontinuities?
> > 2. **The information-theoretic approach is a weak prior.**
> >    - > Our exploration of compressibility was intended to see if a simple, information-theoretic heuristic could serve as a useful prior. The resulting weak and model-dependent correlation is, in itself, a key finding of our paper: it demonstrates that no simple, universal prior across a search space spanning ~10^{365} options exists, thus motivating our more powerful (but complex) reinforcement learning approach.
> >    - This makes sense and aligns with my interpretation of your results! I would try to make the fact that this is a negative result more clear in your paper. In fact, is it even necessary to use the info-theoretic prior as initialization for REOrder? Will row-major initialization work just as well? Its usage for initialization seems to contradict its otherwise weak results and confuses the narrative on how effective this prior actually is.
> > 3. **Interpretability of learned orderings.**
> >    - Your description of back-loading vs front-loading is very interesting and will be a great addition to the paper!
> >
> > I also wanted to re-raise some questions and concerns that were left unanswered:
> >
> > > In Figure 4, why should the patches related to the “keyboard” class move to the end of the patch order? Was that trained on a single image? If not, what’s special about the “keyboard” class such that it dictates the optimal patch order across the entire dataset?
> >
> > > **Missing evaluation in the long sequence regime.** For all experiments in the paper, resolution is set to 224x224 input images, a regime where O(n^2) full self-attention is viable. In this regime, patch ordering is not really a consideration, since full self-attention is permutation invariant. Results would be more persuasive if tested on a truly long sequence regime, such as higher resolution images or videos.

---

> > > ### Author Response · Authors · 2025-08-05
> > >
> > > **ARM Row-Major and Column-Major Scan Ordering**: The performance difference between row-major (57.60% top-1) and column-major (56.96%) inputs for the Mamba/ARM model stems from the anisotropic nature of natural image statistics. Images possess stronger horizontal correlations; therefore, the row-major sequence provides a more informative signal to the model's two row-wise scans than a column-major sequence provides to the two column-wise scans. This inherent data bias gives the row-major ordering a marginal but distinct advantage.
> > >
> > > While a simple column-major sequence is a poor match for ARM's architecture and the statistics of image data, REOrder does not simply learn a slightly better column-major scan. Instead, its reinforcement learning policy learns to create an entirely new, data-specific permutation of the patches that finds a more effective compromise for all four of ARM's competing internal scans.
> > >
> > > REOrder improves the stronger row-major baseline to reaching 58.25%, while it improves the weaker column-major baseline to 58.58%. Our method is able to better utilize the symmetry in ARM's architecture by aligning the underlying data correctly.
> > >
> > > **ARM Hilbert Curve Ordering**: As you noted, a Hilbert curve is designed to maximize 2D spatial locality. Feeding a Hilbert-ordered sequence to ARM's four, fixed scans forces them to process spatially distant patches as if they were adjacent, creating significant discontinuities and degrading performance.
> > >
> > > However, as we discuss above, the ARM architecture is unable to utilize its symmetric scans optimally due to underlying data biases. Therefore, the Hilbert curve test is a fair benchmark for evaluating the adaptability of a learning-based approach. It establishes a challenging, structured-but-suboptimal baseline to test if a method can overcome a fundamental mismatch between the input data's sequence and the model's architecture.
> > >
> > > REOrder, when initialized with a Hilbert curve, achieves a top-1 accuracy on ImageNet-1K of 53.42% compared to 50.41% on training with just a Hilbert curve ordering. This shows that REOrder is capable of learning a fundamentally new permutation that resolves the initial, severe mismatch between the input sequence and the model's architectural bias.
> > >
> > > **Weak Information-Theoretic Prior**: We can frame the compressibility metric as a "negative result" to ensure that the paper's clarity is not impacted. Its primary finding is that no simple, universal heuristic exists, thereby motivating our learning-based approach. Row-major is a reasonable starting point. However, our use of the compressibility prior is intended to be a minor heuristic that nudges the policy search into a potentially promising region of the vast search space from the outset.
> > >
> > > **Keyboard**: This is meant to just be a visually interesting example. We have created visualizations for more classes in ImageNet-1K which demonstrate similar dynamics and will be sure to include them in the supplement.
> > >
> > > **Long sequence regime**: We apologize for not responding to this directly. We kicked off experiments to test this when we read your review. However, because we are training on images that are 5888x5888 (the size at which ViT OOMs) with a batch size, these runs have not finished yet. We do not expect that we will have these results in time for this discussion period.

---

> > > > ### Comment · Reviewer_zkbe · 2025-08-06
> > > >
> > > > **ARM Row-Major and Column-Major Scan Ordering.** This makes sense. Again, though, for my own understanding, shouldn't ARM with row-major and column-major yield theoretically identical results? i.e. the four scan directions for row-major would be:
> > > > - row-major
> > > > - column-major
> > > > - row-backwards-major
> > > > - column-backwards-major
> > > >
> > > > and the four scan directions for column-major would be:
> > > > - column-major + row-major -> column-major
> > > > - column-major + column-major -> row-major
> > > > - column-major + row-backwards-major -> column-backwards-major
> > > > - column-major + column-backwards-major -> row-backwards-major
> > > >
> > > > **ARM Hilbert Curve Ordering**. Understood.
> > > >
> > > > **Weak Information-Theoretic Prior**: Appreciated! This will help clarify the paper's main takeaways.
> > > >
> > > > **Keyboard**. Understood. I would recommend removing or rephrasing this caption
> > > > > We track two patches over the course of the policy curriculum: a keyboard key (light red arrow) and an irrelevant orange beak (dark red arrow). As the policy learns to order patches, we see the patches related to the target class move to the end of the ordering.
> > > >
> > > > since I understand it to be misleading. If I understand correctly, it is just coincidence that the keyboard patches move to the end of the ordering, and the real reason has to do with dataset distribution.
> > > >
> > > > **Long sequence regime**: Acknowledged. It's good to know that these experiments are underway. However, this was a significant concern, so I'm unlikely to bump my overall rating without these results.

---

> > > > > ### Author Response · Authors · 2025-08-06
> > > > >
> > > > > Thank you for your response!
> > > > >
> > > > > **ARM**: You are highlighting a key finding of our paper, that despite theoretical assumptions that there should be identical performance, this is not true in practice! We are the first, to our knowledge, to explore this in depth this for models that do not use full self-attention.
> > > > >
> > > > > **Keyboard**: We do not think that this is a coincidence. In ImageNet-1K, most objects that are used to label the image are in the center of the image. The policy, learned over the entire dataset, learns to move these patches that best suit the inductive biases in the respective architectures. In our visualizations, the class-of-interest often moves to the end.
> > > > >
> > > > > **Long sequence regime**: We apologize for not being able to furnish this result for you. Is there an alternate experiment that would help you gain a better understanding for the behavior? Especially one that may be able to be run by the end of our discussion period.

---

> > > > > > ### Comment · Reviewer_zkbe · 2025-08-06
> > > > > >
> > > > > > **ARM.** Understood, thanks for clarifying!
> > > > > >
> > > > > > **Keyboard.** Yup I agree, and I recommend clarifying this in the final paper, particularly this core fact that’s causing the observed phenomenon.
> > > > > > > In ImageNet-1K, most objects that are used to label the image are in the center of the image.
> > > > > >
> > > > > > **Long sequence regime.** I really appreciate your willingness to explore alternatives! Unfortunately, the experiments that are already underway (but may not finish in time) are the ones that I think may be most impactful for the paper.

---

### Official Review · Reviewer_L4Dr · 2025-06-30

**Clarity:** 3
**Significance:** 4
**Originality:** 3
**Rating:** 4
**Confidence:** 3

**Summary:**

This paper explores how the order of image patches affects the performance of vision models. They especially focus on processing images as long sequences like Transformer-XL, Mamba, and Longformer. While standard vision transformers such as ViT are not sensitive to patch ordering due to full self-attention, newer efficient models introduce inductive biases that make them more sensitive to the sequence of input patches. They show that simply changing the patch order from the common row-major format to alternatives like column-major, spiral, or Hilbert curves can lead to noticeable differences in accuracy depending on the model and dataset. By this experiment, they figured out that the order of patch is non-trivial. To solve this, they propose REOrder, a method that learns task-specific patch orderings using reinforcement learning. REOrder models the patch order as a distribution over permutations using the Plackett-Luce model, and optimizes it with reinforcement learning and Gumbel-top-k sampling. The learned ordering consistently improves classification performance across models and datasets, showing that rethinking patch order is a promising direction for enhancing long-sequence vision models.

**Questions:**

1. It would strengthen the paper to include an analysis or ablation of per-image or per-class adaptive ordering strategies. Such results could help validate whether a single global ordering is sufficient or whether dynamic ordering could further improve performance.

2. Beyond the example shown in Figure 4, could you provide a broader analysis of the learned patch orders across classes or datasets? This would help readers understand what structural or semantic properties the ordering policy captures.

3. It would be helpful to include experiments that evaluate REOrder when initialized from random or very weak patch orderings. This would clarify how much the method relies on having a good prior and whether it remains robust across different starting points.

**Ethical Concerns:**

["NO or VERY MINOR ethics concerns only"]

**Final Justification:**

I believe this work can contribute to further research on the impact of patch ordering, as the results suggest that the order of patches significantly affects performance. In addition, the authors have clearly addressed most of my concerns raised in the earlier review. Therefore, I remain positive about the acceptance of this paper.

**Limitations:**

Yes

**Paper Formatting Concerns:**

I didn't find any major formatting issues.

**Quality:**

3

**Strengths And Weaknesses:**

Strengths

1. Insightful approach: The paper convincingly identifies and analyzes a practical inductive bias in long-sequence vision models, sensitivity to patch ordering, which has been largely underexplored. This sheds light on a subtle but impactful limitation in common transformer variants and challenges the assumption that input ordering is inconsequential. It's a fresh perspective that could influence how future vision architectures are designed and trained.

2. Generalizable Solution: The proposed method, REOrder, is easy to be plugged into many transformer-based models without architectural changes. The use of a Plackett-Luce distribution with REINFORCE and Gumbel-top-k sampling is well-motivated and makes the optimization over permutations tractable. It’s a rare case where a relatively simple reinforcement learning setup yields consistent and non-trivial gains across diverse architectures and datasets.

3. Clarity of paper and their experiences: The experiments are quite comprehensive covering different patch orderings, multiple models. Furthermore, their supplementary further show their training details which further demonstrates their effectiveness and robustness.

Weakness

1. Global ordering limits flexibility: REOrder learns a single patch permutation shared across all inputs for each model-dataset pair. This approach may not capture per-image structural differences, especially in diverse datasets like ImageNet. The paper does not discuss this limitation or possible dynamic extensions.

2. Lack of interpretability or analysis of learned orderings: Beyond a single illustrative example, the paper does not analyze what kinds of patterns the learned orderings reflect or why they help. There is little insight into how orderings vary across classes, datasets, or models, which limits understanding and generalizability.

3. Limited analysis on the impact of patch ordering priors: While the paper uses six predefined patch orderings as priors, it lacks a deeper analysis of how the choice of initial ordering affects REOrder’s performance. In some cases, REOrder appears to achieve substantial gains even when starting from a weak prior (e.g., Snake for Mamba with 'Functional Map of the world' dataset), but this is not explicitly discussed. Furthermore, there is no comparison against random patch orderings, which would help clarify how much the method relies on having a reasonable starting point. Without this, it’s difficult to assess the robustness and sensitivity of REOrder to the choice of prior.

---

> ### Author Rebuttal · Authors · 2025-07-30
>
> We thank the reviewer for their detailed and constructive review of our work. We appreciate that you recognize the value in how our RL-based approach can lead to positive outcomes across a diverse set of architectures and datasets. The weaknesses you’ve identified are excellent points that give us a clear pathway to improving our paper. We address these in depth below:
>
> **Global ordering limits flexibility**: We agree with the reviewer that learning a single global ordering for an entire dataset is a limitation and that dynamic, per-image ordering is an exciting direction for future work. Our primary goal in this paper was to first prove conclusively that (1) patch ordering effects are real and significant in modern long-sequence models, and (2) that reinforcement learning is a viable strategy for discovering superior orderings. By learning a single, global permutation, we deliberately isolated the effect of the ordering itself, providing a clean and stable experimental setup without the confounding variance of a more complex, dynamic policy network. One way to break global ordering-associated limitations is to modify REOrder to use a small policy network that takes image patch embeddings as input and generates the ordering logits $z$ dynamically. We will update our Limitations and Future Work section accordingly.
>
> **Lack of interpretability**: We thank the reviewer for this point, which prompted us to conduct a deeper analysis of the patterns learned by REOrder. The “keyboard” example in Figure 4 suggested a link to semantic importance, but we agree a broader analysis is needed. Our new investigation reveals that REOrder learns to align the spatial structure of a dataset with the specific inductive biases of different model architectures.
>
> Our analysis is grounded in two key observations: (1) long-sequence models are sensitive to input order, and (2) ImageNet-1K and Functional Map of the World (FMoW) exhibit a strong center bias, where the most semantically critical content is usually in the middle of the image. To investigate this, we quantified how REOrder repositions patches from the image’s central region relative to their initial ordering. This can reveal whether the learned policy prioritizes central patches by moving them to the beginning (“front-loading”) or the end (“back-loading”) of the sequence. We discovered two distinct, model-dependent strategies:
>
> 1. Mamba prefers to front-load central tokens. REOrder systematically moved central patches towards the start of the sequence (on average, a shift of +6.8 positions). This “front-loading” strategy intuitively aligns with Mamba's recurrent state-space mechanism. By processing the most salient patches first, the model establishes a strong initial state ($h_t$​) that informs the processing of the entire sequence. This allows the model to build a robust understanding of the image’s core subject from the outset.
>
> 2. Longformer and Transformer-XL prefer to back-load central tokens. Both models learned to shift central patches towards the __end__ of the sequence (average shifts of -20.4 and -2.2 positions, respectively). Transformer-XL’s recurrent memory can summarize information from earlier, less critical patches. By positioning central patches later in the sequence, these tokens can utilize information from the entire image rather than only from half of it. Comparatively, Longformer’s sliding-window attention, which accumulates context over time, would provide greater information for patches near the end. This could maximize the contextual information available for the local and global attention mechanisms when classifying objects.
>
> This contrast between “front-loading” for Mamba and “back-loading” for Transformer-XL/Longformer on the same ImageNet dataset demonstrates that REOrder is not finding a single, trivial ordering. Instead, it is learning sophisticated, model-specific policies that exploit how each architecture processes information. We will add this detailed analysis and discussion to the final version.
>
> **Limited choice of priors**: This is a great point to highlight. Since the search space of possible permutations is prohibitively large (~10^{365} for 196 patches), we provide an information-theoretic strategy such that we can start with a strong prior for our learning process. To directly test the importance of this initialization and the robustness of our RL algorithm, we have performed the ablation study you suggested. We ran additional experiments on ImageNet-1K with Transformer-XL to compare REOrder's performance when initialized with structured priors versus a completely random permutation. The Top-1 accuracy results are as follows:
> * Baseline (row-major): 60.81
> * REOrder with a row-major prior: 61.40
> * REOrder with a column-major prior: 62.78
> * REOrder with a completely random prior: 53.10
>
> These results demonstrate that the choice of prior is critical. While our RL algorithm can explore the permutation space, its ability to find a high-performing solution is significantly hampered by a random, unstructured start. This finding validates our core approach: using an information-theoretic strategy to select a sensible prior is crucial for making the learning process tractable and effective.

---

> > ### Comment · Reviewer_L4Dr · 2025-08-05
> >
> > Thank you for the detailed and thoughtful responses. I appreciate the additional analysis and experiments you conducted based on the feedback.
> > In particular, adding a section on interpretability in the final version will help address some of the ambiguity present in the current draft.
> > I’m satisfied with the clarifications and have no further questions at this time.
> > After discussing with the other reviewers, I will consider the final score.

---

### Official Review · Reviewer_C815 · 2025-07-02

**Clarity:** 4
**Significance:** 2
**Originality:** 3
**Rating:** 4
**Confidence:** 4

**Summary:**

This manuscript presents an interesting question; whether the ordering of patches affects the performance of variants of ViT.
It is well known that the vanilla self-attention is permutation-invariant. However, some variants of ViT do not feed all the low-level patches. This would make a limited window of tokens like limited perceptive field.

The authors empirically showed that the ordering matters in Longformer, Mamba and Trransformer-XL.
To alleviate (ot to utilize) this phenomenon, The authors propose RL-based solution.

**Questions:**

It is not clear why RL-based method is not effective for Longformer.
It would be better to explore nontent-aware dynamic re-ordering method.

**Ethical Concerns:**

["NO or VERY MINOR ethics concerns only"]

**Final Justification:**

Based on the authors's rebuttal to reviews, I am willing to increase the scores.

**Limitations:**

This method is effective Mamba models.
Though, the authors argue that there is a hidden opportunity of improving performance using better reordering, the   RL method did not add up extra gain over fixed ordering in ViT-variant.

**Paper Formatting Concerns:**

The formatting and writing is excellent.

**Quality:**

3

**Strengths And Weaknesses:**

This paper presents an interesting problem and empirically shows the significance of ordering in some models.
The RL-based approach shows significant improvement in Mamba model.
The evaluation is clean and somewhat through.

However, the proposed method basically work only for Mamba.  In case of Mamba, as well discussed in Section 3.3, the architecture is not permutation-invariant from the beginning. It is also well know that Mamba is sequence-dependent.
It may not be proper to the ordering of patched in Mamba and ViT-variant together.

Based on the limited benefit/cost of the proposed method in ViT-variants, I don't think this method would be adopted to ViT-based models.

---

> ### Author Rebuttal · Authors · 2025-07-30
>
> We thank the reviewer for their positive assessment of our paper’s quality, clarity, and the interesting nature of the problem. We appreciate the opportunity to clarify some key aspects of our work and address the valid questions raised. The central thesis of our paper is that many modern, efficient vision models, regardless of whether they are based on attention approximations, recurrence, or state-space models, sacrifice the permutation equivariance of the original Vision Transformer. Our goal was to investigate this shared sensitivity to patch ordering across different architectural families. We deliberately chose Longformer, Transformer-XL, and Mamba not as “ViT-variants” vs. “Mamba”, but as representative examples of three distinct strategies for achieving efficiency, all of which, as we demonstrate in Section 3.3, become sensitive to input order as a result. We address your questions in depth below:
>
> **A perceived lack of improvement in performance for ViT-variants**: Our results show that REOrder provides significant and consistent performance gains for Transformer-XL as well. Specifically, on ImageNet-1K, REOrder improved Transformer-XL's accuracy by 1.50% over the fixed Hilbert curve ordering and 1.09% over the spiral ordering. On the Functional Map of the World dataset, REOrder improved Transformer-XL's best-performing baseline (column-major) by an additional 1.10%. While the gains in performance for T-XL and Longformer are not as drastic as they are for Mamba, they are statistically significant across the board.
>
> **Cost of REOrder**: REOrder adds as many parameters as there are patches. For a standard 224x224 image with 16x16 patches, this is 196 parameters. This increases parameter count for a ViT-Large-equivalent model by ~$2.0\times10^{-4}$%. This makes REOrder an extremely efficient method that adds negligible runtime and memory overhead to a standard supervised learning setup.
>
> **Effectiveness for Longformer**: This is an important nuance that speaks to the model's architecture. As we note in our conclusion, Longformer's design, which combines a sliding-window attention with global attention tokens, is a closer approximation of full self-attention compared to Transformer-XL or Mamba. Because it was the least susceptible to patch ordering changes among the three models to begin with (as seen in Figure 2), it is expected (and we observe that) that REOrder has less room for improvement. This finding supports our overall thesis that the degree of ordering sensitivity is tied to the degree to which a model deviates from full, permutation-equivariant self-attention.
>
> **Exploring a content-aware mechanism**: We appreciate this suggestion. REOrder is a content-aware method insofar that the learned patch ordering is specific to a model/dataset pair. A permutation is then sampled based on these logits. This means the final patch order is determined by the learned relevance of different parts of the image content for the specific task, making it inherently content-aware. Interesting future work can include conditioning the RL-policy on __every__ image rather than a dataset at large. We will ensure this aspect of our method is described more clearly in the main text.

---

> > ### Comment · Reviewer_C815 · 2025-08-04
> > **Good explanations**
> >
> > Thank you for your explanations. Though I am not sure about the claim that REOrder is a content-aware method insofar, based on the explanations on my questions and other reviews, I would like to increase the core.

---

### Official Review · Reviewer_bd6c · 2025-07-03

**Clarity:** 3
**Significance:** 2
**Originality:** 3
**Rating:** 4
**Confidence:** 3

**Summary:**

This paper addresses the problem that the design of patch ordering in autoregressive vision models has a significant impact on performance. Based on the hypothesis that conventional fixed patch ordering does not guarantee optimality, the authors propose a method to dynamically determine the ordering using reinforcement learning. By applying the proposed method to the Mamba architecture, for instance, the authors demonstrate significant performance improvements on the FMoW dataset.

**Questions:**

1 Can you please explain why ViT's accuracy (37.5%) on ImageNet-1k is significantly lower and how this affects your assessment of the proposed method's effectiveness?

2 Could you elaborate on the reasons for selecting Transformer-XL, Longformer, and Mamba as baselines and the significance of each comparison?

3 In light of recent developments such as the 2D extension of Mamba and bidirectional processing, what do you think is the practical standing of this research?

4 Have you verified the performance of the proposed method on out-of-distribution data or downstream tasks to rule out the possibility of overlearning to a particular order? Or do you have any future plans to do so?

**Ethical Concerns:**

["NO or VERY MINOR ethics concerns only"]

**Final Justification:**

This paper can be evaluated for providing a new perspective on the fundamental problem of patch ordering in vision models using patch tokens. It also proposes a method for reinforcement learning to find appropriate patch ordering. Taking into consideration the comments and discussions from other reviewers, I will maintain our original score.

**Limitations:**

yes

**Quality:**

3

**Strengths And Weaknesses:**

Strengths

1. The description is clear and well-structured throughout the paper.

2. The issue of the effect of patch order on autoregressive image models is novel and interesting in that it has not been fully explored.

3. The proposed method provides a framework for dynamically optimizing the patch order based on information-theoretic initialization and reinforcement learning, and is applicable to implementations. In particular, the approach of designing the initial order in terms of information compression is unique and suggestive.

Weaknesses

1. Concerns about insufficient training in the experimental setting:
Figure 2 shows only performance differences and omits absolute accuracy. On the other hand, line 194 of the text states that ViT achieves only 37.5% in Top-1 Accuracy for ImageNet-1k. This is extremely low compared to the performance of standard ViT (in the upper 70% range in the original ViT paper), probably due to the lack of convergence caused by the short training period of 100 epochs. In evaluating the effectiveness of the proposed method, a comparison under more fully trained conditions would be desirable, and it is difficult to draw strong conclusions from the current experimental results. This concern is similar for the other architecture, the other data (FMoW).

2 Basis for selection of baseline models is unclear:
Transformer-XL, Longformer, and Mamba are selected, but the reasons for their selection and their relationship to this method are not clearly presented. breaking Permutation-invariance While there are many variants of Transformer, why these three were chosen and what comparative insights can be gained from them?

3. Ambiguous positioning with the technical context:
 Although the proposed method focuses on sequential image models, many efficient and high-performance approaches have already been proposed for application to image tasks using Mamba and other methods, such as 2D extensions of SSM and bi-directional processing. In such a technological context, it is somewhat unclear what practical and theoretical contributions this study has to offer.

4. Overlearning Concerns for Patch Order:
 Even if the proposed method is effective in a particular pre-training task, it is unclear whether it will generalize to out-of-distribution data and downstream tasks. This is a major practical concern. In particular, additional experiments on out-of-distribution generalization performance and transition learning tasks are desirable to eliminate the possibility that the patch order is overfitting the training data.

---

> ### Author Rebuttal · Authors · 2025-07-30
>
> Thank you for providing a detailed and insightful review for our paper. We appreciate your recognition of the novelty of exploring the effects of for-granted architectural decisions such as patch ordering, especially as the community moves increasingly to long-sequence models. Below, we answer the questions you posed in your review, present some extra experimental details, and attempt to clarify any misconceptions that may have arisen in our writeup:
>
> **Top-1 Accuracy on ImageNet-1K**: Our goal with this work was to isolate the effects of patch ordering. To do so, we simplified the training recipe to its basics and kept it static across all runs: 100 epochs, $1\times10^{-4}$ learning rate, no augmentations, and a cosine learning rate decay. For this rebuttal, we increased the training length to 300 epochs. For T-XL, this improved performance to ~62% top-1 accuracy on ImageNet-1K. For T-XL with REOrder (row-major prior), this improved performance to ~63.5%. The ordering in our results from epoch 100 to epoch 300 did not change, nor did the relative magnitude of the results.
>
> The original ViT paper achieved 77.91% top-1 accuracy on ImageNet-1K, but their training and evaluation setup is drastically different, which explains their higher performance. Dosovitskiy et. al. first pre-train on ImageNet at 224px resolution for 300 epochs, and then fine-tune on ImageNet at 384px resolution. In comparison, our work is trained fully-supervised from random initialization for 100 epochs (300 epochs in the above experiment). The gap in performance to the ViT paper is then largely explained by pre-training and high-resolution inference. Training for longer maintains the ordering of our results.
>
> **Selection of T-XL, Longformer, and ARM (Mamba)**: We selected Transformer-XL, Longformer, and Mamba (specifically the ARM implementation) as they represent three distinct and influential architectural approaches to handling long-sequence data. Transformer-XL introduces a segment-level recurrence mechanism and a novel relative positional encoding scheme. This design allows the model to capture dependencies beyond a fixed length by reusing hidden states from previous segments, effectively creating a memory. Longformer addresses the quadratic complexity of the Transformer's self-attention mechanism by employing a combination of a sliding window (local) attention and a global attention mechanism. Mamba introduces a structured state-space model that operates with linear-time complexity and constant memory scaling. Each of these models were SOTA on benchmarks such as WikiText-103, One Billion Word, etc. at the time of release and are still near-SOTA today.
>
> Our work shows that regardless of the long-sequence modeling philosophy, patch order effects have a significant impact on downstream task performance.
>
> **Improvement to Mamba and bi-direction scan**: This is a good point, and we apologize that our paper was not more clear about this point.The Mamba implementation we utilize throughout our work is ARM (Autoregressive Pretraining with Mamba in Vision) [0], a model that incorporates multi-directional scanning. As we detail in our Preliminaries (Section 3.3) and Methods (Section 4), ARM is specifically designed for vision tasks and extends the base Mamba architecture significantly. Specifically, for each layer, ARM processes the input by running four parallel, causal scans in different directions (left, right, up, down) and then combines their outputs. This architecture inherently includes the bidirectional (e.g., forward and backward) and multi-axis (e.g., row-wise and column-wise) processing that you mentioned. Our work shows that even with these multi-directional scanning mechanisms, the model's performance remains highly sensitive to the initial 1-D patch ordering. We show that the challenge of patch ordering is not a relic of simpler, unidirectional models but a persistent issue in state-of-the-art architectures like ARM. REOrder is able to drastically improve performance even for these SOTA SSM architectures.
>
> **Pre-training and out-of-distribution performance**: REOrder is fully-supervised with respect to the images, while the patch ordering is learned without supervision with an RL setup. The core premise of our work is that the optimal patch ordering is not universal, but is instead dependent on the specific pairing of a model’s architecture and a dataset’s intrinsic properties. REOrder is designed to discover and exploit this pairing. However, your suggestion of pre-training brings up an interesting idea for future work: in lieu of an information-theoretic prior, a model and the REOrder policy would be pre-trained together on a large-scale dataset. The resulting learned policy would then serve as a prior for natural images in future tasks. During fine-tuning on a new downstream task, both the model weights and the policy logits could be updated, allowing the learned prior patch ordering to further adapt to the specific requirements of the new task.
>
> [0] https://arxiv.org/abs/2406.07537v1

---

> ### Comment · Reviewer_bd6c · 2025-08-04
>
> Your kind response has helped me understand several points. Thank you very much. However, I still have some questions regarding the following points.
>
> **Top-1 Accuracy on ImageNet-1K**: I believe that the additional experiments you conducted have provided evidence that supports the usefulness of your proposed method. As you explained, we should take into account the fact that fine-tuning to high resolution was not performed. However, as I understand it, ViT typically achieves accuracy in the 70% range on ImageNet-1K even with standard settings. Therefore, I remain somewhat concerned about the performance gap that still exists. I suspect this difference may be due to the lack of data augmentation (CutMix, for instance), but I do not believe that there are any factors that restrict the use of these techniques with the proposed method.
>
> **Improvement to Mamba and bidirectional scan**: I apologize for my lack of understanding regarding this point. My question was to ask how specifically superior the proposed method is compared to existing Mamba-type models optimized for image tasks, such as [1] [2]. I understand that the training conditions in this study were not necessarily optimized for performance, but It is unfortunate that we were unable to quantitatively understand how much the proposed method could potentially improve SOTA performance on ImageNet-1K. On the other hand, given the additional experiments mentioned above, I understand that sufficient qualitative improvement can be expected.
>
> In any case, the review-discussion process has deepened my understanding of this research, and I view it favorably overall. I will carefully consider the final score, taking into account discussions with other reviewers.
>
> [1] Liu et al., VMamba: Visual State Space Model, NeurIPS 2024
>
> [2] Zhu et al., Vision Mamba: Efficient Visual Representation Learning with Bidirectional State Space Model, ICML 2024

---

> > ### Author Response · Authors · 2025-08-05
> >
> > **Top-1 Accuracy on ImageNet-1K**: Prior work from the ViT team [0] indicates that with the original setup (pre-training on ImageNet-1K, inference at 384px), they are able to achieve 67.1% top-1 accuracy at 300 epochs. Our 300 epoch run on T-XL is able to achieve 62% top-1 accuracy, which is not far off from the ViT reproduction!
> >
> > We believe the difference in performance here is largely due to pre-training. We are not using pre-trained weights and are training from scratch so that we can isolate the ordering effects directly.
> >
> > We regret not being able to fully answer your question due to our limited computational budget. We will aim to train our models for longer in time for the camera-ready deadline.
> >
> > **Comparison to VMamba and Vision Mamba**: We use ARM because it is SOTA when compared to prior work like VMamba and Vision Mamba. While these works make different architectural decisions from ARM, we expect that our results will hold. We can run these experiments and add them to the supplement.
> >
> > [0] Better plain ViT baselines for ImageNet-1k. Beyer et. al. https://arxiv.org/pdf/2205.01580

---

> > > ### Comment · Reviewer_bd6c · 2025-08-08
> > >
> > > Thank you for your additional explanation.
> > >
> > > I understand that the baseline performance of 67.1% in [0] does not use any of the currently standard augmentation techniques such as MixUp. This is exactly what I pointed out last time.
> > >
> > > Thank you also for explaining why the comparison with other architectures such as VMamba is lacking.

---

### Official Review · Reviewer_pVqq · 2025-07-04

**Clarity:** 3
**Significance:** 2
**Originality:** 4
**Rating:** 4
**Confidence:** 4

**Summary:**

The paper investigates the effect of the patch order on the performance of transformer-influenced image classification models that are not equivariant to permutations of patches (together with positional embeddings).

The paper presents the hypothesis that the standard row-major patch ordering might not be the optimal one and proposes a policy learning algorithm that learns the patch ordering in parallel with the classifier (for some number of epochs before freezing the policy). The policy learning uses the REINFORCE algorith with a policy model with the number of parameters corresponding to the number of patches (Plackett-Luce model).

The paper tests one equivariant architecture, ViT, and 3 non equivariant ones, Linformer, Transformer-XL, and Mamba (ARM). First, we see that changing the patch order of the model degrades the performance of non-equivariant models. For example, ImageNet-1k top-1 accuracy drops most on the Mamba model (around 6%) when a trained model is tested on different patch orders (row-major to column-major, spiral, ...).

Training the model with the proposed algorithm improves the performance with respect to the sub-optimal patch orderings, and the learned model performs similarly to the model trained with the row-mayor patch ordering.

**Questions:**

**Questions:**

1. Please address weakness points 1 - 3.
2. Why is the patch size selected to be 32×32?

I hope that the authors will be able to address all the important weaknesses. I apologize for any errors on my part.

**Ethical Concerns:**

["NO or VERY MINOR ethics concerns only"]

**Final Justification:**

I anticipate that the paper will be updated based on the discussion, including various clarifications and the already performed experiment that shows the importance of patch-ordering evolution during training. Therefore, I have increased my score to borderline accept.

**Limitations:**

The paper comments on some limitations in the appendix. Additional limitations seem to be the lack of experiments with fixed previously learned patch orderings and low baseline accuracies.

**Paper Formatting Concerns:**

No formatting issues noticed.

**Quality:**

3

**Strengths And Weaknesses:**

**Strengths**

1. The source code for reproducing the experiments is provided will publicly available.
2. The paper is written well, has clear hypotheses, and presents an original investigation.
3. The research questions of the paper make sense.

**Weaknesses**

1. Validation experiments could be improved.
	1. If I understand correctly, the main paper does not include experiments with supervised classification training from the start with a learned patch ordering and with a random ordering, but only concurrently with permutation learning. Experiments singling out the effect of the learned permutation would be valuable.
	2. An experiment with a random patch ordering is included in the appendix but not mentioned in the main paper. I think that the substantially worse performance on a random patch ordering is a valuable data point.
2. Clarity could be improved.
	1. What is $M_\text{TXL}$?
	2. I understand that all the Mamba experiments use ARM (Ren at al.), but I am not sure. Please consider clarifying it.
	3. Figure 2:
		1. Is the y-axis relative change or absolute (percentage points)? ()
		2. Does Figure 2 use a single training run?
		3. L190: "We evaluated top-1 accuracy on validation sets, estimating the Standard Error of the Mean (SEM) using a non-parametric bootstrap method with 2,000 resamples." What is the sizes of the samples? Is the standard error of the mean 0 if the size equals the whole validation dataset?
	4. Figure 3. Why is the ImageNet-1k top-1-accuracy of the ViT (ViT-B/32?) baseline (Row) only around 38%?
3. The discussion about REOrder experiments (Section 6.4 and Conclusion) could be improved.
	1. I think that there is too much focus on the results of improving patch orderings that are worse than the default row-major ordering (Hilbert curve and diagonal), while what matters most is the improvement over row-major.
	2. How does the validation accuracy vary over different training runs?
	3. The percentages can be ambiguous. I suggest saying "pp" (percentage points).
4. Presentation could be improved (easy to address).
	3. Not all appendix sections are referenced in the main paper, while some are quite valuable (for example, Appendix A).
	4. Some more model details could be presented in the main paper for better information availability and sanity-checking. For example, it takes some searching to find out that "ViT" is ViT-Base and that the patch size is 32×32 (if I am correct). Also, is the Mamba in the experiments ARM?
	5. Figure 3: The y-axes of the ViT experiments is a bit broken. Some numbers occur on multiple ticks.
	6. Missing details:
		1. For how many epochs is the patch ordering learned? (What is the value of $M$?)
		2. What is the shape of the temperature schedule?

Minor suggestions:
- The notation seems inconsistent: some matrices are not in bold.

---

> ### Author Rebuttal · Authors · 2025-07-30
>
> Thank you for your time and thoroughness in your review! Your feedback is incredibly valuable. We appreciate that you find our work interesting and find the writing to be well-done. Based on your review, we have conducted some additional experiments and re-organized the paper for clarity. We discuss each point in depth below:
>
> **Including more supervised learning experiments for the baselines**: We previously ran these experiments but under-estimated the importance of including them in the main body of the paper. We will put these in the main body of the paper for the camera-ready version. You suggested running two experiments: (1) picking a static, random permutation once at the start of training, and (2) taking a previously-learned ordering and statically training with it. In addition to what you suggested, we added an additional experiment: (3) For every iteration during training, pick a random permutation ordering.
>
> 1. tests if any arbitrary permutation produces better or worse results than row-major or REOrder.
> 2. tests whether the learned permutation is what matters most, or the process of guided exploration added by REOrder
> 3. tests whether if unguided exploration is better than guided exploration
>
> For the results below, we use Transformer-XL Base with all of the same parameters as described in the paper and report the top-1 accuracy on ImageNet-1K. We first repeat results from the paper for row-major and REOrder with a row-major prior, and then provide results for (1), (2), and (3).
> * Row major: 60.80
> * Row major with REOrder (row prior): 61.40
> * (1) We pick a random permutation ahead of training and then use that for the entire learning process: 53.10
> * (2) Take the best learned ordering from (b), fix that, and use that to train from the beginning: 53.07
> * (3) Random permutation for every batch during training: 50.23
>
> All together, these results add to the narrative of our paper: simply picking random patch orderings is not sufficient (and therefore, any one random ordering will also perform poorly). The structured exploration provided by REOrder is necessary to improve performance. A novel outcome of this exploration is that simply using a learned patch ordering is not sufficient for gains in performance; the guided exploration process introduced by REOrder is necessary for improved performance.
>
> **Paper clarity**: We will re-organize the paper for experimental clarity. To answer your specific questions, $M_{TXL}$ is the length of the memory for T-XL, which was 128 in our experiments). We use ARM as our Mamba implementation of choice and mention this on L150 in the Preliminaries section. However, we will re-label our figures to be explicit about ARM. For all experiments, we use a patch size of 16. For visualization purposes only, Figure 6 in the Appendix shows a patch size of 32. We will be sure to address the clarity comments you highlighted in our revision.
>
> **Figure related comments**: In general, our reported numbers (and Figure 2) reports percentage points. We will clarify this explicitly. We report results using multiple training runs. There are instances in which our figures have broken y-axes, such as Figure 3. These are issues caused due to rounding and will be rectified!
>
> **Top-1 Accuracy on ImageNet-1K**: Our goal with this work was to isolate the effects of patch ordering. To do so, we simplified the training recipe to its basics and kept it static across all runs: 100 epochs, $1\times10^{-4}$ learning rate, no augmentations, and a cosine learning rate decay. For this rebuttal, we increased the training length to 300 epochs. For T-XL, this improved performance to ~62% top-1 accuracy on ImageNet-1K. For T-XL with REOrder (row-major prior), this improved performance to ~63.5%. The ordering in our results from epoch 100 to epoch 300 did not change, nor did the relative magnitude of the results.
>
> The original ViT paper achieved 77.91% top-1 accuracy on ImageNet-1K, but their training and evaluation setup is drastically different, which explains their higher performance. Dosovitskiy et. al. first pre-train on ImageNet at 224px resolution for 300 epochs, and then fine-tune on ImageNet at 384px resolution. In comparison, our work is trained fully-supervised from random initialization for 100 epochs (300 epochs in the above experiment). The gap in performance to the ViT paper is then largely explained by pre-training and high-resolution inference. Training for longer maintains the ordering of our results.

---

> > ### Comment · Reviewer_pVqq · 2025-08-02
> > **Good clarification, valuable new experiments**
> >
> > Thank you! Your response has clarified some things. I have also considered the responses to the other reviews. Now I am likely to increase my score.
> >
> > **Low Top-1 Accuracy on ImageNet-1K.** The response has clarified the issue. I think that the low baseline accuracy due to less involved training makes the results less relevant and is the main weakness of the paper.
> >
> > **Importance of patch-ordering evolution during training.**
> >
> > > simply using a learned patch ordering is not sufficient for gains in performance; the guided exploration process introduced by REOrder is necessary for improved performance.
> >
> > Thank you! I find these results very interesting. Please check whether you have to update some claims that emphasize the importance of the final patch ordering rather than the training process with REOrder.
> >
> > If you have time and think it makes sense, please also consider the following experiment for further checking the importance of the final patch order:
> > 1. Record the sequence of all patch orderings during training.
> > 2. Perform another training run, but, instead of running REOrder, use the recorded sequence of patch orderings in reverse.
> >
> > **Front-loading and back-loading.** I don't understand the explanations of why front- and back-loading happens. Could you please check or clarify why Mamba benefits from positioning the central tokens at the beginning to produce a "strong" initial state that informs the processing of the entire sequence, but Transformer-XL's recurrent memory benefits from positioning them at the end of the sequence. Why is it not the other way around?

---

> > > ### Author Response · Authors · 2025-08-05
> > >
> > > **Low Top-1 Accuracy**: The reviewer is correct that the models are not fully optimized. As we demonstrated in our previous response, the ordering and magnitude of our results does not change given increased training time and higher top-1 accuracies. To ensure that results are comparable against literature, we will run full 300-epoch training runs for the camera-ready version of the paper.
> > >
> > > **Replay policy**: We are working on launching these experiments! We're not sure if they will finish in time given limited access to compute, but we will report back as soon as we have something.
> > >
> > > **Interpretability of front- and back-loading**: We are working on better mechanisms by which to get interpretable results for the observed behaviors. Ultimately, the expressed behaviors are a complex amalgamation of attention, architecture, and the policy dynamics. Our current best understanding is that Mamba processes tokens in strict sequence and updates its hidden state at each step. Placing central tokens first lets it form a strong semantic representation early, which it then refines with peripheral patches. Conversely, Transformer-XL uses a causal attention mask and limited memory (128 tokens for 196 patches plus CLS). Tokens at the end attend to more previous tokens and are more likely to be stored in memory. Positioning central tokens last ensures they integrate information from the entire image and remain accessible for later steps.
> > >
> > > Please let us know if this intuition helps with your understanding. Ultimately, the interpretability work may be useful future work that is independent work of this paper.

---

> > > > ### Comment · Reviewer_pVqq · 2025-08-07
> > > >
> > > > I appreciate your effort in improving the evaluation. Also, thank you for the clarification about front- and back-loading. The claim (hypothesis) makes more sense to me now, but I would need to invest some more time to be able to check it thoroughly.
> > > >
> > > > I anticipate that the paper will be updated based on the discussion, including various clarifications and the already performed experiment that shows the importance of patch-ordering evolution during training. Therefore, I will increase my score to borderline accept.

---

### Decision · Program_Chairs · 2025-09-17

**Decision:**

Accept (poster)

**Comment:**

The paper makes an empirical observation that despite transformers being generally permutation-invariant over the input tokens, vision transformers, are sensitive to the image tokenization process (patchification). That is, the advanced vision transformers use techniques (such as for computational efficiency) that renders them dependent on the ordering of the patches. This is due to the 2D nature of images where no flattening into a 1D sequence would be perfect. The paper, then, proposes a process to optimize the order (permutation) of the patches based on the task (loss) at hand using REINFORCE. They demonstrate significant improvement in the classification accuracy of ImageNet and FMoW.

The paper received five expert reviews. The authors provided a thorough rebuttal which was attended and discussed by the reviewers. Eventually, all reviewers leaned towards acceptance, highlighting the originality and importance of the observation, the relevance of the proposed method, and the significance of the achieved results. There remained some concerns on the lack of dynamic per-image ordering or low baseline results on which the reviewers did not agree in the AC-reviewers discussion.

The AC does not see any outstanding major concern with the paper. Vision transformers are becoming increasingly common in computer vision research, particularly those versions with efficiency tricks that make them potentially more sensitive to ordering. The paper makes an important observation and study of on the often-neglected importance of the ordering which reveals significant dependency of the optimal ordering to the task/dataset and architectures. This, by itself, is worth disseminating. It further proposes a method to improve the initial ordering and mitigate the dependence on it which shows the promise in such future direction. Based on these two significant contributions and the unanimous reviewers' recommendation, the AC suggests acceptance.